# Convergence of Adversarial Training in Overparametrized Neural Networks

Ruiqi Gao[1,*]    Tianle Cai[1,*]    Haochuan Li[2]    Liwei Wang[3]    Cho-Jui Hsieh[4]    Jason D. Lee[5]

[1]School of Mathematical Sciences, Peking University
[2]Department of EECS, Massachusetts Institute of Technology
[3]Key Laboratory of Machine Perception, MOE, School of EECS, Peking University
[4]Department of Computer Science, University of California, Los Angeles
[5]Department of Electrical Engineering, Princeton University

## Abstract

Neural networks are vulnerable to adversarial examples, i.e. inputs that are imperceptibly perturbed from natural data and yet incorrectly classified by the network. Adversarial training [31], a heuristic form of robust optimization that alternates between minimization and maximization steps, has proven to be among the most successful methods to train networks to be robust against a pre-defined family of perturbations. This paper provides a partial answer to the success of adversarial training, by showing that it converges to a network where the surrogate loss with respect to the the attack algorithm is within $\epsilon$ of the optimal robust loss. Then we show that the optimal robust loss is also close to zero, hence adversarial training finds a robust classifier. The analysis technique leverages recent work on the analysis of neural networks via Neural Tangent Kernel (NTK), combined with motivation from online-learning when the maximization is solved by a heuristic, and the expressiveness of the NTK kernel in the $\ell_\infty$-norm. In addition, we also prove that robust interpolation requires more model capacity, supporting the evidence that adversarial training requires wider networks.

## 1   Introduction

Recent studies have demonstrated that neural network models, despite achieving human-level performance on many important tasks, are not robust to adversarial examples—a small and human imperceptible input perturbation can easily change the prediction label [44, 22]. This phenomenon brings out security concerns when deploying neural network models to real world systems [20]. In the past few years, many defense algorithms have been developed [23, 43, 30, 28, 39] to improve the network's robustness, but most of them are still vulnerable under stronger attacks, as reported in [3]. Among current defense methods, adversarial training [31] has become one of the most successful methods to train robust neural networks.

To obtain a robust network, we need to consider the "robust loss" instead of a regular loss. The robust loss is defined as the maximal loss within a neighborhood around the input of each sample, and minimizing the robust loss under empirical distribution leads to a min-max optimization problem. Adversarial training [31] is a way to minimize the robust loss. At each iteration, it (approximately) solves the inner maximization problem by an attack algorithm $\mathcal{A}$ to get an adversarial sample, and then runs a (stochastic) gradient-descent update to minimize the loss on the adversarial samples. Although adversarial training has been widely used in practice and hugely improves the robustness of neural networks in many applications, its convergence properties are still unknown. It is unclear

whether a network with small robust error exists and whether adversarial training is able to converge to a solution with minimal adversarial train loss.

In this paper, we study the convergence of adversarial training algorithms and try to answer the above questions on over-parameterized neural networks. We consider width-$m$ neural networks both for the setting of deep networks with $H$ layers, and two-layer networks for some additional analysis. Our contributions are summarized below.

- For an $H$-layer deep network with ReLU activations, and an arbitrary attack algorithm, when the width $m$ is large enough, we show that projected gradient descent converges to a network where the surrogate loss with respect to the attack $\mathcal{A}$ is within $\epsilon$ of the optimal robust loss (Theorem 4.1). The required width is *polynomial* in the depth and the input dimension.

- For a two-layer network with smooth activations, we provide a proof of convergence, where the projection step is not required in the algorithm (Theorem 5.1).

- We then consider the expressivity of neural networks w.r.t. robust loss (or robust interpolation). We show when the width $m$ is sufficiently large, the neural network can achieve optimal robust loss $\epsilon$; see Theorems 5.2 and C.1 for the precise statement. By combining the expressivity result and the previous bound of the loss over the optimal robust loss, we show that adversarial training finds networks of small robust training loss (Corollary 5.1 and Corollary C.1).

- We show that the VC-Dimension of the model class which can *robustly* interpolate any $n$ samples is lower bounded by $\Omega(nd)$ where $d$ is the dimension. In contrast, there are neural net architectures that can interpolate $n$ samples with only $O(n)$ parameters and VC-Dimension at most $O(n \log n)$. Therefore, the capacity required for robust learning is higher.

## 2 Related Work

**Attack and Defense**    Adversarial examples are inputs that are slightly perturbed from a natural sample and yet incorrectly classified by the model. An adversarial example can be generated by maximizing the loss function within an $\epsilon$-ball around a natural sample. Thus, generating adversarial examples can be viewed as solving a constrained optimization problem and can be (approximately) solved by a projected gradient descent (PGD) method [31]. Some other techniques have also been proposed in the literature including L-BFGS [44], FGSM [22], iterative FGSM [26] and C&W attack [12], where they differ from each other by the distance measurements, loss function or optimization algorithms. There are also studies on adversarial attacks with limited information about the target model. For instance, [13, 24, 8] considered the black-box setting where the model is hidden but the attacker can make queries and get the corresponding outputs of the model.

Improving the robustness of neural networks against adversarial attacks, also known as defense, has been recognized as an important and unsolved problem in machine learning. Various kinds of defense methods have been proposed [23, 43, 30, 28, 39], but many of them are based on obfuscated gradients which does not really improve robustness under stronger attacks [3]. As an exception, [3] reported that the adversarial training method developed in [31] is the only defense that works even under carefully designed attacks.

**Adversarial Training**    Adversarial training is one of the first defense ideas proposed in earlier papers [22]. The main idea is to add adversarial examples into the training set to improve the robustness. However, earlier work usually only adds adversarial example once or only few times during the training phase. Recently, [31] showed that adversarial training can be viewed as solving a min-max optimization problem where the training algorithm aims to minimize the robust loss, defined as the maximal loss within a certain $\epsilon$-ball around each training sample. Based on this formulation, a clean adversarial training procedure based on PGD-attack has been developed and achieved state-of-the-art results even under strong attacks. This also motivates some recent research on gaining theoretical understanding of robust error [9, 40]. Also, adversarial training suffers from slow training time since it runs several steps of attacks within one update, and several recent works are trying to resolve this issue [41, 53]. From the theoretical perspective, a recent work [46]

considers to quantitatively evaluate the convergence quality of adversarial examples found in the inner maximization and therefore ensure robustness. [51] consider generalization upper and lower bounds for robust generalization. [29] improves the robust generalization by data augmentation with GAN. [21] considers to reduce the optimization of min-max problem to online learning setting and use their results to analyze the convergence of GAN. In this paper, our analysis for adversarial is quite general and is not restricted to any specific kind of attack algorithm.

**Global convergence of Gradient Descent**   Recent works on the over-parametrization of neural networks prove that when the width greatly exceeds the sample size, gradient descent converges to a global minimizer from random initialization [27, 18, 19, 1, 55]. The key idea in the earlier literature is to show that the Jacobian w.r.t. parameters has minimum singular value lower bounded, and thus there is a global minimum near every random initialization, with high probability. However for the robust loss, the maximization cannot be evaluated and the Jacobian is not necessarily full rank. For the surrogate loss, the heuristic attack algorithm may not even be continuous and so the same arguments cannot be utilized.

**Certified Defense and Robustness Verification**   In contrast to attack algorithms, neural network verification methods [48, 47, 54, 42, 14, 38] tries to find upper bounds of the robust loss and provide certified robustness measurements. Equipped with these verification methods for computing upper bounds of robust error, one can then apply adversarial training to get a network with certified robustness. Our analysis in Section 4 can also be extended to certified adversarial training.

# 3   Preliminaries

## 3.1   Notations

Let $[n] = \{1, 2, \ldots, n\}$. We use $\mathcal{N}(\mathbf{0}, \mathbf{I})$ to denote the standard Gaussian distribution. For a vector $\mathbf{v}$, we use $\|\mathbf{v}\|_2$ to denote the Euclidean norm. For a matrix $\mathbf{A}$ we use $\|\mathbf{A}\|_F$ to denote the Frobenius norm and $\|\mathbf{A}\|_2$ to denote the spectral norm. We use $\langle \cdot, \cdot \rangle$ to denote the standard Euclidean inner product between two vectors, matrices, or tensors. We let $O(\cdot)$, $\Theta(\cdot)$ and $\Omega(\cdot)$ denote standard Big-O, Big-Theta and Big-Omega notations that suppress multiplicative constants.

## 3.2   Deep Neural Networks

Here we give the definition of our deep fully-connected neural networks. For the convenience of proof, we use the same architecture as defined in [1].[2] Formally, we consider a neural network of the following form.

Let $\mathbf{x} \in \mathbb{R}^d$ be the input, the fully-connected neural network is defined as follows: $\mathbf{A} \in \mathbb{R}^{m \times d}$ is the first weight matrix, $\mathbf{W}^{(h)} \in \mathbb{R}^{m \times m}$ is the weight matrix at the $h$-th layer for $h \in [H]$, $\mathbf{a} \in \mathbb{R}^{m \times 1}$ is the output layer, and $\sigma(\cdot)$ is the ReLU activation function.[3]   The parameters are $\mathbf{W} = (\text{vec}\{\mathbf{A}\}^\top, \text{vec}\{\mathbf{W}^{(1)}\}^\top, \cdots, \text{vec}\{\mathbf{W}^{(H)}\}^\top, \mathbf{a}^\top)^\top$. However, without loss of generality, during training we will fix $\mathbf{A}$ and $\mathbf{a}$ once initialized, so later we will refer to $\mathbf{W}$ as $\mathbf{W} = (\text{vec}\{\mathbf{W}^{(1)}\}^\top, \cdots, \text{vec}\{\mathbf{W}^{(H)}\}^\top)^\top$. The prediction function is defined recursively:

$$
\begin{aligned}
\mathbf{x}^{(0)} &= \mathbf{A}\mathbf{x} \\
\overline{\mathbf{x}}^{(h)} &= \mathbf{W}^{(h)}\mathbf{x}^{(h-1)}, \quad h \in [H] \\
\mathbf{x}^{(h)} &= \sigma\left(\overline{\mathbf{x}}^{(h)}\right), \quad h \in [H] \\
f(\mathbf{W}, \mathbf{x}) &= \mathbf{a}^\top \mathbf{x}^{(H)},
\end{aligned} \tag{1}
$$

where $\overline{\mathbf{x}}^{(h)}$ and $\mathbf{x}^{(h)}$ are the feature vectors before and after the activation function, respectively. Sometimes we also denote $\overline{\mathbf{x}}^{(0)} = \mathbf{x}^{(0)}$.

We use the following initialization scheme: Each entry in $\mathbf{A}$ and $\mathbf{W}^{(h)}$ for $h \in [H]$ follows the i.i.d. Gaussian distribution $\mathcal{N}(0, \frac{2}{m})$, and each entry in $\mathbf{a}$ follows the i.i.d. Gaussian distribution $\mathcal{N}(0, 1)$. As we mentioned, we only train on $\mathbf{W}^{(h)}$ for $h \in [H]$ and fix $\mathbf{a}$ and $\mathbf{A}$. For a training set $\{\mathbf{x}_i, y_i\}_{i=1}^n$, the loss function is denoted $\ell : (\mathbb{R}, \mathbb{R}) \mapsto \mathbb{R}$, and the (non-robust) training loss is $L(\mathbf{W}) = \frac{1}{n} \sum_{i=1}^n \ell(f(\mathbf{W}, \mathbf{x}_i), y_i)$. We make the following assumption on the loss function:

**Assumption 3.1** (Assumption on the Loss Function). *The loss $\ell(f(\mathbf{W}, \mathbf{x}), y)$ is Lipschitz, smooth, convex in $f(\mathbf{W}, \mathbf{x})$ and satisfies $\ell(y, y) = 0$.*

### 3.3 Perturbation and the Surrogate Loss Function

The goal of adversarial training is to make the model robust in a neighbor of each datum. We first introduce the definition of the perturbation set function to determine the perturbation at each point.

**Definition 3.1** (Perturbation Set). *Let the input space be $\mathcal{X} \subset \mathbb{R}^d$. The perturbation set function is $\mathcal{B} : \mathcal{X} \to \mathcal{P}(\mathcal{X})$, where $\mathcal{P}(\mathcal{X})$ is the power set of $\mathcal{X}$. At each data point $\mathbf{x}$, $\mathcal{B}(\mathbf{x})$ gives the perturbation set on which we would like to guarantee robustness. For example, a commonly used perturbation set is $\mathcal{B}(\mathbf{x}) = \{\mathbf{x}' : \|\mathbf{x}' - \mathbf{x}\|_2 \leq \delta\}$. Given a dataset $\{\mathbf{x}_i, y_i\}_{i=1}^n$, we say that the perturbation set is compatible with the dataset if $\overline{\mathcal{B}(\mathbf{x}_i)} \cap \overline{\mathcal{B}(\mathbf{x}_j)} \neq \phi$ implies $y_i = y_j$. In the rest of the paper, we will always assume that $\mathcal{B}$ is compatible with the given data.*

Given a perturbation set, we are now ready to define the perturbation function that maps a data point to another point inside its perturbation set. We note that the perturbation function can be quite general including the identity function and any adversarial attack[4]. Formally, we give the following definition.

**Definition 3.2** (Perturbation Function). *A perturbation function is defined as a function $\mathcal{A} : \mathcal{W} \times \mathbb{R}^d \to \mathbb{R}^d$, where $\mathcal{W}$ is the parameter space. Given the parameter $\mathbf{W}$ of the neural network (1), $\mathcal{A}(\mathbf{W}, \mathbf{x})$ maps $\mathbf{x} \in \mathbb{R}^d$ to some $\mathbf{x}' \in \mathcal{B}(\mathbf{x})$ where $\mathcal{B}(\mathbf{x})$ refers to the perturbation set defined in Definition 3.1.*

Without loss of generality, throughout Section 4 and 5, we will restrict our input $\mathbf{x}$ as well as the perturbation set $\mathcal{B}(\mathbf{x})$ within the surface of the unit ball $\mathcal{S} = \{\mathbf{x} \in \mathbb{R}^d : \|\mathbf{x}\|_2 = 1\}$.

With the definition of perturbation function, we can now define a large family of loss functions on the training set $\{\mathbf{x}_i, y_i\}_{i=1}^n$. We will show this definition covers the standard loss used in empirical risk minimization and the robust loss used in adversarial training.

**Definition 3.3** (Surrogate Loss Function). *Given a perturbation function $\mathcal{A}$ defined in Definition 3.2, the current parameter $\mathbf{W}$ of a neural network $f$, and a training set $\{\mathbf{x}_i, y_i\}_{i=1}^n$, we define the surrogate loss $L_{\mathcal{A}}(\mathbf{W})$ on the training set as*

$$L_{\mathcal{A}}(\mathbf{W}) = \frac{1}{n} \sum_{i=1}^n \ell(f(\mathbf{W}, \mathcal{A}(\mathbf{W}, \mathbf{x}_i)), y_i).$$

It can be easily observed that the standard training loss $L(\mathbf{W})$ is a special case of surrogate loss function when $\mathcal{A}$ is the identity. The goal of adversarial training is to minimize the *robust loss*, i.e. the surrogate loss when $\mathcal{A}$ is the strongest possible attack. The formal definition is as follows:

**Definition 3.4** (Robust Loss Function). *The robust loss function is defined as*

$$L_*(\mathbf{W}) := L_{\mathcal{A}^*}(\mathbf{W})$$

*where*

$$\mathcal{A}^*(\mathbf{W}, \mathbf{x}_i) = \underset{\mathbf{x}_i' \in \mathcal{B}(\mathbf{x}_i)}{\operatorname{argmax}} \ell(f(\mathbf{W}, \mathbf{x}_i'), y_i).$$

## 4 Convergence Results of Adversarial Training

We consider optimizing the surrogate loss $L_{\mathcal{A}}$ with the perturbation function $\mathcal{A}(\mathbf{W}, \mathbf{x})$ defined in Definition 3.2, which is what adversarial training does given any attack algorithm $\mathcal{A}$. In this section,

we will prove that for a neural network with sufficient width, starting from the initialization $\mathbf{W}_0$, after certain steps of projected gradient descent within a convex set $B(R)$, the loss $L_{\mathcal{A}}$ is provably upper-bounded by the best minimax robust loss in this set

$$\min_{\mathbf{W} \in B(R)} L_*(\mathbf{W}),$$

where

$$B(R) = \left\{ \mathbf{W} : \left\| \mathbf{W}^{(h)} - \mathbf{W}_0^{(h)} \right\|_F \leq \frac{R}{\sqrt{m}}, h \in [H] \right\}. \tag{2}$$

Denote $\mathcal{P}_{B(R)}$ as the Euclidean projection to the convex set $B(R)$. Denote the parameter $\mathbf{W}$ after the $t$-th iteration as $\mathbf{W}_t$, and similarly $\mathbf{W}_t^{(h)}$. For each step in adversarial training, projected gradient descent takes an update

$$\mathbf{V}_{t+1} = \mathbf{W}_t - \alpha \nabla_{\mathbf{W}} L_{\mathcal{A}}(\mathbf{W}_t),$$
$$\mathbf{W}_{t+1} = \mathcal{P}_{B(R)}(\mathbf{V}_{t+1}),$$

where

$$\nabla_{\mathbf{W}} L_{\mathcal{A}}(\mathbf{W}) = \frac{1}{n} \sum_{i=1}^n l' \left( f(\mathbf{W}, \mathcal{A}(\mathbf{W}, \mathbf{x}_i)), y_i \right) \nabla_{\mathbf{W}} f(\mathbf{W}, \mathcal{A}(\mathbf{W}, \mathbf{x}_i)),$$

and the derivative $\ell'$ stands for $\frac{\partial \ell}{\partial f}$, *the gradient $\nabla_{\mathbf{W}} f$ is with respect to the first argument* $\mathbf{W}$.

Specifically, we have the following theorem.

**Theorem 4.1** (Convergence of Projected Gradient Descent for Optimizing Surrogate Loss). *Given $\epsilon > 0$, suppose $R = \Omega(1)$, and $m \geq \max\left(\Theta\left(\frac{R^9 H^{16}}{\epsilon^7}\right), \Theta(d^2)\right)$. Let the loss function satisfy Assumption 3.1.[5] If we run projected gradient descent based on the convex constraint set $B(R)$ with stepsize $\alpha = O\left(\frac{\epsilon}{mH^2}\right)$ for $T = \Theta\left(\frac{R^2}{m\epsilon\alpha}\right) = \Omega\left(\frac{R^2 H^2}{\epsilon^2}\right)$ steps, then with high probability we have*

$$\min_{t=1,\cdots,T} L_{\mathcal{A}}(\mathbf{W}_t) - L_*(\mathbf{W}_*) \leq \epsilon, \tag{3}$$

*where $\mathbf{W}_* = \arg\min_{\mathbf{W} \in B(R)} L_*(\mathbf{W})$.*

**Remark.** *Recall that $L_{\mathcal{A}}(\mathbf{W})$ is the loss suffered with respect to the perturbation function $\mathcal{A}$. This means, for example, if the adversary uses the projected gradient ascent algorithm, then the theorem guarantees that projected gradient ascent cannot successfully attack the learned network. The stronger the attack algorithm is during training, the stronger the guaranteed surrogate loss becomes.*

**Remark.** *The value of $R$ depends on the approximation capability of the network, i.e. the greater $R$ is, the less $L_*(\mathbf{W}_*)$ will be, thus affecting the overall bound on $\min_t L_{\mathcal{A}}(\mathbf{W}_t)$. We will elaborate on this in the next section, where we show that for $R$ independent of $m$ there exists a network of small adversarial training error.*

### 4.1 Proof Sketch

Our proof idea utilizes the same high-level intuition as [1, 27, 18, 55, 10, 11] that near the initialization the network is linear. However, unlike these earlier works, the surrogate loss neither smooth, nor semi-smooth so there is no Polyak gradient domination phenomenon to allow for the global geometric contraction of gradient descent. In fact due to the the generality of perturbation function $\mathcal{A}$ allowed, the surrogate loss is not differentiable or even continuous in $\mathbf{W}$, and so the standard analysis cannot be applied. Our analysis utilizes two key observations. First the network $f(\mathbf{W}, \mathcal{A}(\mathbf{W}, \mathbf{x}))$ is still smooth w.r.t. the first argument[6], and is close to linear in the first argument near initialization, which is shown by directly bounding the Hessian w.r.t. $\mathbf{W}$. Second, the perturbation function $\mathcal{A}$ can be treated as an adversary providing a worst-case loss function $\ell_{\mathcal{A}}(f, y)$ as done in online learning. However, online learning typically assumes the sequence of losses is convex, which is not the case here. We make a careful decoupling of the contribution to non-convexity from the first argument and the worst-case contribution from the perturbation function, and then we can prove that gradient descent succeeds in minimizing the surrogate loss. The full proof is in Appendix A.

# 5 Adversarial Training Finds Robust Classifier

Motivated by the optimization result in Theorem 4.1, we hope to show that there is indeed a robust classifier in $B(R)$. To show this, we utilize the connection between neural networks and their induced Reproducing Kernel Hilbert Space (RKHS) via viewing networks near initialization as a random feature scheme [15, 16, 25, 2]. Since we only need to show the existence of a network architecture that robustly fits the training data in $B(R)$ and neural networks are at least as expressive as their induced kernels, we may prove this via the RKHS connection. The strategy is to first show the existence of a robust classifier in the RKHS, and then show that a sufficiently wide network can approximate the kernel via random feature analysis. The approximation results of this section will be, in general, exponential in dimension dependence due to the known issue of $d$-dimensional functions having exponentially large RKHS norm [4], so only offer *qualitative guidance on existence of robust classifiers*.

Since deep networks contain two-layer networks as a sub-network, and we are concerned with expressivity, we focus on the local expressivity of two-layer networks. We write the standard two-layer network in the suggestive way[7] (where the width $m$ is an even number)

$$f(\mathbf{W}, \mathbf{x}) = \frac{1}{\sqrt{m}} \left( \sum_{r=1}^{m/2} a_r \sigma(\mathbf{w}_r^\top \mathbf{x}) + \sum_{r=1}^{m/2} a_r' \sigma(\bar{\mathbf{w}}_r^\top \mathbf{x}) \right), \tag{4}$$

and initialize as $\mathbf{w}_r \sim \mathcal{N}(0, \mathbf{I}_d)$ i.i.d. for $r = 1, \cdots, \frac{m}{2}$, and $\bar{\mathbf{w}}_r$ is set to be equal to $\mathbf{w}_r$, $a_r$ is randomly drawn from $\{1, -1\}$ and $a_r' = -a_r$. Similarly, we define the set $B(R) = \{\mathbf{W} : \|\mathbf{W} - \mathbf{W}_0\|_F \leq R\}$[8] for $\mathbf{W} = (\mathbf{w}_1, \cdots, \mathbf{w}_{m/2}, \bar{\mathbf{w}}_1, \cdots, \bar{\mathbf{w}}_{m/2})$, $\mathbf{W}_0$ being the initialization of $\mathbf{W}$, and fix all $a_r$ after initialization.

To make things cleaner, we will use a smooth activation function $\sigma(\cdot)$ throughout this section[9], formally stated as follows.

**Assumption 5.1** (Smoothness of Activation Function). *The activation function $\sigma(\cdot)$ is smooth, that is, there exists an absolute constant $C > 0$ such that for any $z, z' \in \mathbb{R}$*

$$|\sigma'(z) - \sigma'(z')| \leq C|z - z'|.$$

Prior to proving the approximation results, we would like to first provide a version of convergence theorem similar to Theorem 4.1, but for this two-layer setting. It is encouraged that the reader can read Appendix B for the proof of the following Theorem 5.1 first, since it is relatively cleaner than that of the deep setting but the proof logic is analogous.

**Theorem 5.1** (Convergence of Gradient Descent *without Projection* for Optimizing Surrogate Loss for Two-layer Networks). *Suppose the loss function satisfies Assumption 3.1 and the activation function satisfies Assumption 5.1. With high probability, using the two-layer network defined above, for any $\epsilon > 0$, if we run gradient descent with step size $\alpha = O(\epsilon)$, and if $m = \Omega\left(\frac{R^4}{\epsilon^2}\right)$, we have*

$$\min_{t=1,\cdots,T} L_\mathcal{A}(\mathbf{W}_t) - L_*(\mathbf{W}_*) \leq \epsilon, \tag{5}$$

*where $\mathbf{W}_* = \min_{\mathbf{W} \in B(R)} L_*(\mathbf{W})$ and $T = \Theta(\frac{\sqrt{m}}{\alpha})$.*

**Remark.** *Compared to Theorem 4.1, we do not need the projection step for this two-layer theorem. We believe using a smooth activation function can also eliminate the need of the projection step in the deep setting from a technical perspective, and from a practical sense we conjecture that the projection step is not needed anyway.*

Now we're ready to proceed to the approximation results, i.e. proving that $L_*(\mathbf{W}_*)$ is also small, and combined with Equation (5) we can give an absolute bound on $\min_t L_\mathcal{A}(\mathbf{W}_t)$. For the reader's convenience, we first introduce the Neural Tangent Kernel (NTK) [25] w.r.t. our two-layer network.

**Definition 5.1** (NTK [25]). *The NTK with activation function $\sigma(\cdot)$ and initialization distribution $\mathbf{w} \sim \mathcal{N}(0, \mathbf{I}_d)$ is defined as $K_\sigma(\mathbf{x}, \mathbf{y}) = \mathbb{E}_{\mathbf{w} \sim \mathcal{N}(0, \mathbf{I}_d)} \langle \mathbf{x}\sigma'(\mathbf{w}^\top \mathbf{x}), \mathbf{y}\sigma'(\mathbf{w}^\top \mathbf{y}) \rangle$.*

For a given kernel $K$, there is a reproducing kernel Hilbert space (RKHS) introduced by $K$. We denote it as $\mathcal{H}(K)$. We refer the readers to [36] for an introduction of the theory of RKHS.

We formally make the following assumption on the universality of NTK.

**Assumption 5.2** (Existence of Robust Classifier in NTK). *For any $\epsilon > 0$, there exists $f \in \mathcal{H}(K_\sigma)$, such that $|f(\mathbf{x}'_i) - y_i| \leq \epsilon$, for every $i \in [n]$ and $\mathbf{x}'_i \in \mathcal{B}(\mathbf{x}_i)$.*

Also, we make an additional assumption on the activation function $\sigma(\cdot)$:

**Assumption 5.3** (Lipschitz Property of Activation Function). *The activation function $\sigma(\cdot)$ satisfies $|\sigma'(z)| \leq C, \forall z \in \mathbb{R}$ for some constant $C$.*

Under these assumptions, by applying the strategy of approximating the infinite situation by finite sum of random features, we can get the following theorem:

**Theorem 5.2** (Existence of Robust Classifier near Initialization). *Given data set $\mathcal{D} = \{(\mathbf{x}_i, y_i)\}_{i=1}^n$ and a compatible perturbation set function $\mathcal{B}$ with $\mathbf{x}_i$ and its allowed perturbations taking value on $\mathcal{S}$, for the two-layer network defined in (4), if Assumption 3.1, 5.1, 5.2, 5.3 hold, then for any $\epsilon, \delta > 0$, there exists $R_{\mathcal{D},\mathcal{B},\epsilon}$ such that when the width $m$ satisfies $m = \Omega\left(\frac{R_{\mathcal{D},\mathcal{B},\epsilon}^4}{\epsilon^2}\right)$, with probability at least $0.99$ over the initialization there exists $\mathbf{W}$ such that*

$$L_*(\mathbf{W}) \leq \epsilon \text{ and } \mathbf{W} \in B(R_{\mathcal{D},\mathcal{B},\epsilon}).$$

Combining Theorem 5.1 and 5.2 we finally know that

**Corollary 5.1** (Adversarial Training Finds a Network of Small Robust Training Loss). *Given data set on the unit sphere equipped with a compatible perturbation set function and an associated perturbation function $\mathcal{A}$, which also takes value on the unit sphere. Suppose Assumption 3.1, 5.1, 5.2, 5.3 are satisfied. Then there exists a $R_{\mathcal{D},\mathcal{B},\epsilon}$ which only depends on dataset $\mathcal{D}$, perturbation $\mathcal{B}$ and $\epsilon$, such that for any 2-layer fully connected network with width $m = \Omega(\frac{R_{\mathcal{D},\mathcal{B},\epsilon}^4}{\epsilon^2})$, if we run gradient descent with stepsize $\alpha = O(\epsilon)$ for $T = \Theta(\frac{R_{\mathcal{D},\mathcal{B},\epsilon}^2}{\epsilon\alpha})$ steps, then with probability $0.99$,*

$$\min_{t=1,\cdots,T} L_\mathcal{A}(\mathbf{W}_t) \leq \epsilon. \tag{6}$$

**Remark 5.1.** *We point out that Assumption 5.2 is rather general and can be verified for a large class of activation functions by showing their induced kernel is universal as done in [32]. Also, here we use an implicit expression of the radius $B_{\mathcal{D},\mathcal{B},\epsilon}$, but the dependence on $\epsilon$ can be calculated under specific activation function with or without the smoothness assumptions. As an example, using quadratic ReLU as activation function, we solve the explicit dependency on $\epsilon$ in Appendix C.2 that doesn't rely on Assumption 5.2.*

Therefore, adversarial training is guaranteed to find a robust classifier under a given attack algorithm when the network width is sufficiently large.

# 6 Capacity Requirement of Robustness

In this section, we will show that in order to achieve adversarially robust interpolation (which is formally defined below), one needs more capacity than just normal interpolation. In fact, empirical evidence have already shown that to reliably withstand strong adversarial attacks, networks require a significantly larger capacity than for correctly classifying benign examples only [31]. This implies, in some sense, that using a neural network with larger width is necessary.

Let $\mathcal{S}_\delta = \{(\mathbf{x}_1, \cdots, \mathbf{x}_n) \in (\mathbb{R}^d)^n : \|\mathbf{x}_i - \mathbf{x}_j\|_2 > 2\delta\}$ and $\mathcal{B}_\delta(\mathbf{x}) = \{\mathbf{x}' : \|\mathbf{x}' - \mathbf{x}\|_2 \leq \delta\}$, where $\delta$ is a constant. We consider datasets in $\mathcal{S}_\delta$ and use $\mathcal{B}_\delta$ as the perturbation set function in this section.

We begin with the definition of the interpolation class and the robust interpolation class.

**Definition 6.1** (Interpolation class). *We say that a function class $\mathcal{F}$ of functions $f : \mathbb{R}^d \to \{1, -1\}$ is an $n$-interpolation class[10], if the following is satisfied:*

$$\forall (\mathbf{x}_1, \cdots, \mathbf{x}_n) \in \mathcal{S}_\delta, \forall (y_1, \cdots, y_n) \in \{\pm 1\}^n,$$
$$\exists f \in \mathcal{F}, \ s.t. \ f(\mathbf{x}_i) = y_i, \forall i \in [n].$$

**Definition 6.2** (Robust interpolation class). *We say that a function class $\mathcal{F}$ is an $n$-robust interpolation class, if the following is satisfied:*

$$\forall (\mathbf{x}_1, \cdots, \mathbf{x}_n) \in \mathcal{S}_\delta, \forall (y_1, \cdots, y_n) \in \{\pm 1\}^n,$$
$$\exists f \in \mathcal{F}, s.t. f(\mathbf{x}_i') = y_i, \forall \mathbf{x}_i' \in \mathcal{B}_\delta(\mathbf{x}_i), \forall i \in [n].$$

We will use the VC-Dimension of a function class $\mathcal{F}$ to measure its complexity. In fact, as shown in [6] (Equation(2)), for neural networks there is a tight connection between the number of parameters $W$, the number of layers $H$ and their VC-Dimension

$$\Omega(HW \log(W/H)) \leq \text{VC-Dimension} \leq O(HW \log W).$$

In addition, combining with the results in [52] (Theorem 3) which shows the existence of a 4-layer neural network with $O(n)$ parameters that can interpolate any $n$ data points, i.e. an $n$-interpolation class, we have that an $n$-interpolation class can be realized by a fixed depth neural network with VC-Dimension upper bound

$$\text{VC-Dimension} \leq O(n \log n). \tag{7}$$

For a general hypothesis class $\mathcal{F}$, we can evidently see that when $\mathcal{F}$ is an $n$-interpolation class, $\mathcal{F}$ has VC-Dimension at least $n$. For a neural network that is an $n$-interpolation class, without further architectural constraints, this lower bound of its VC-dimension is tight up to logarithmic factors as indicated in Equation (7). However, we show that for a robust-interpolation class we will have a much larger VC-Dimension lower bound:

**Theorem 6.1.** *If $\mathcal{F}$ is an $n$-robust interpolation class, then we have the following lower bound on the VC-Dimension of $\mathcal{F}$*

$$VC\text{-}Dimension \geq \Omega(nd), \tag{8}$$

*where $d$ is the dimension of the input space.*

For neural networks, Equation (8) shows that any architecture that is an $n$-robust interpolation class should have VC-Dimension at least $\Omega(nd)$. Compared with Equation (7) which shows an $n$-interpolation class can be realized by a network architecture with VC-Dimension $O(n \log n)$, we can conclude that robust interpolation by neural networks needs more capacity, so increasing the width of neural network is indeed in some sense necessary.

# 7   Discussion on Limitations and Future Directions

This work provides a theoretical analysis of the empirically successful adversarial training algorithm in the training of robust neural networks. Our main results indicate that adversarial training will find a network of low robust surrogate loss, even when the maximization is computed via a heuristic algorithm such as projected gradient ascent. However, there are still some limitations with our current theory, and we also feel our results can lead to several thought-provoking future work, which is discussed as follows.

*Removal of projection.* It is also natural to ask whether the projection step can be removed, as it is empirically unnecessary and also unnecessary for our two-layer analysis. We believe using smooth activations might resolve this issue from a technical perspective, although practically it seems the projection step in the algorithm is unnecessary in any case.

*Generalizing to different attacks.* Firstly, our current guarantee of the surrogate loss is based on the same perturbation function as that used during training. It is natural to ask that whether we can ensure

the surrogate loss is low with respect to a larger family of perturbation functions than that used during training.

*Exploiting structures of network and data.* Same as the recent proof of convergence on overparameterized networks in the non-robust setting, our analysis fails to further incorporate useful network structures apart from being sufficiently wide, and as a result increasing depth can only hurt the bound. It would be interesting to provide finer analysis based on additional assumptions on the alignment of the network structure and data distribution.

*Improving the approximation bound.* On the expressivity side, the current argument utilizes that a neural net restricted to a local region can approximate its induced RKHS. Although the RKHS is universal, they do not avoid the curse of dimensionality (see Appendix C.2). However, we believe in reality, the required radius of region $R$ to achieve robust approximation is not as large as the theorem demands. So an interesting question is whether the robust expressivity of neural networks can adapt to structures such as low latent dimension of the data mechanism [17, 50], thereby reducing the approximation bound.

*Capacity requirement of robustness and robust generalization.* Apart from this paper, there are other works supporting the need for capacity including the perspective of network width [31], depth [49] and computational complexity [35]. It is argued in [51] that robust generalization is also harder using Rademacher complexity. In fact, it appears empirically that robust generalization is even harder than robust training. It is observed that increasing the capacity, though benifiting the dacay of training loss, has much less effect on robust generalization. There are also other factors behind robust generalization, like the number of training data [40]. The questions about robust generalization, as well as to what extent capacity influences it, are still subject to much debate.

The above are several interesting directions of further improvement to our current result. In fact, many of these questions are largely unanswered even for neural nets in the non-robust setting, so we leave them to future work.

## 8 Acknowlegements

We acknowlegde useful discussions with Siyu Chen, Di He, Runtian Zhai, and Xiyu Zhai. RG and TC are partially supported by the elite undergraduate training program of School of Mathematical Sciences in Peking University. LW acknowledges support by Natioanl Key R&D Program of China (no. 2018YFB1402600), BJNSF (L172037). JDL acknowledges support of the ARO under MURI Award W911NF-11-1-0303, the Sloan Research Fellowship, and NSF CCF #1900145.

## Footnotes

*Joint first author.

[2]We only consider the setting when the network output is scalar. However, it is not hard to extend out results to the setting of vector outputs.

[3]We assume intermediate layers are square matrices of size $m$ for simplicity. It is not difficult to generalize our analysis to rectangular weight matrices.

[4]It is also not hard to extend our analysis to perturbation functions involving randomness.

[5] We actually didn't use the assumption $\ell(y, y) = 0$ in the proof, so common loss functions like the cross-entropy loss works in this theorem. Also, with some slight modifications, it is possible to prove for other loss functions including the square loss.

[6] It is not jointly smooth in $\mathbf{W}$, which is part of the subtlety of the analysis.

[7]This makes $f(\mathbf{W}, \mathbf{x}) = 0$ at initialization, which helps eliminate some unnecessary technical nuisance.

[8]Note that we have taken out the term $\frac{1}{\sqrt{m}}$ explicitly in the network expression for convenience, so in this section there is a difference of scaling by a factor of $\sqrt{m}$ from the $\mathbf{W}$ used in the previous section.

[9]Similar approximation results also hold for other activation functions like ReLU.

[10]Here we let the classification output be $\pm 1$, and a usual classifier $f$ outputting a number in $\mathbb{R}$ can be treated as $\text{sign}(f)$ here.

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
