[Supplementary Material]

# Convergence of Adversarial Training in Overparametrized Neural Networks (Appendix)

## A   Proof of the Convergence Result for Deep Nets in Section 4

We first present some useful notations and lemmas for this part. Denote the diagonal matrix $\mathbf{D}^{(h)} = \mathbf{D}^{(h)}(\mathbf{W}, \mathbf{x})$ as $\mathbf{D}^{(h)} = \text{diag}\left(\mathbb{I}(\overline{\mathbf{x}}^{(h)} \geq 0)\right)$ for $h \in [H]$, where $\mathbb{I}$ is the entry-wise indicator function. Sometimes we also denote $\mathbf{D}^{(0)} = \mathbf{I}$ which is the identity matrix. Therefore, the neural network has the following formula

$$f(\mathbf{W}, \mathbf{x}) = \mathbf{a}^\top \mathbf{D}^{(H)} \mathbf{W}^{(H)} \cdots \mathbf{D}^{(1)} \mathbf{W}^{(1)} \mathbf{x}^{(0)}$$

and the gradient w.r.t. $\mathbf{W}^{(h)}$ is

$$f'^{(h)}(\mathbf{W}, \mathbf{x}) = \left(\mathbf{x}^{(h-1)} \mathbf{a}^\top \mathbf{D}^{(H)} \mathbf{W}^{(H)} \cdots \mathbf{D}^{(h)}\right)^\top, \qquad h \in [H]. \tag{1}$$

First, we will restate some basic results at initialization.

**Lemma A.1.** *If $m \geq d$, with probability $1 - O(H)e^{-\Omega(m)}$ at initialization, we have $\|\mathbf{A}\|_2 = O(1)$, $\left\|\mathbf{W}^{(h)}\right\|_2 = O(1), \forall h \in [H]$, and $\|\mathbf{a}\|_2 = O(\sqrt{m})$.*

*Proof.* This is a well-known result of the $l_2$-norm of Gaussian random matrices (Corollary 5.35 in [8]), which states that for a matrix $\mathbf{M} \in \mathbb{R}^{a \times b}$ with i.i.d. standard Gaussian entries, with probability $1 - 2e^{-t^2/2}$ we have $\|\mathbf{M}\|_2 \leq \sqrt{a} + \sqrt{b} + t$. Combined with the scaling of $\mathbf{A}, \mathbf{W}^{(h)}$, and $\mathbf{a}$, we easily know that each of $\|\mathbf{A}\|_2 = O(1)$, $\left\|\mathbf{W}^{(h)}\right\|_2 = O(1)$, and $\|\mathbf{a}\|_2 = O(\sqrt{m})$ holds with probability $1 - e^{-\Omega(m)}$, and we obtain our result by taking the union event. $\square$

**Lemma A.2.** *For any fixed input $\mathbf{x} \in \mathcal{S}$, with probability $1 - O(H)e^{-\Omega(m/H)}$ over the randomness of initialization, we have for every $h \in \{0, \ldots, H\}$, $\left\|\overline{\mathbf{x}}^{(h)}\right\|_2 \in [2/3, 4/3]$ and $\left\|\mathbf{x}^{(h)}\right\|_2 \in [2/3, 4/3]$ at initialization.*

*Proof.* This is a restatement of Lemma 7.1 in [1] taking the number of data $n = 1$. $\square$

**Lemma A.3.** *If $m = \Omega(H \log H)$, for any fixed input $\mathbf{x} \in \mathcal{S}$, with probability $1 - e^{-\Omega(m/H)}$, at initialization we have for every $h \in [H]$,*

$$\left\|\mathbf{a}^\top \mathbf{D}^{(H)} \mathbf{W}^{(H)} \cdots \mathbf{D}^{(h)} \mathbf{W}^{(h)}\right\|_2 = O(\sqrt{mH}).$$

*Proof.* Note that $\left\|\mathbf{a}^\top \mathbf{D}^{(H)} \mathbf{W}^{(H)} \cdots \mathbf{D}^{(h)} \mathbf{W}^{(h)}\right\|_2 \leq \left\|\mathbf{a}^\top\right\|_2 \left\|\mathbf{D}^{(H)}\right\|_2 \left\|\mathbf{W}^{(H)} \cdots \mathbf{D}^{(h)} \mathbf{W}^{(h)}\right\|_2$ and $\left\|\mathbf{D}^{(H)}\right\|_2 \leq 1$. This lemma then becomes a direct consequence of Lemma A.1 and Lemma 7.3(a) in [1] with number of data $n = 1$. $\square$

Our general idea is that within the local region (where $R = \Omega(1)$)

$$B(R) = \{\mathbf{W} : \left\|\mathbf{W}^{(h)} - \mathbf{W}_0^{(h)}\right\|_F \leq \frac{R}{\sqrt{m}}, \forall h \in [H]\}$$

the gradient $f'^{(h)}(\mathbf{W})$ remains stable over $\mathbf{W}$ when $\mathbf{x}$ is fixed, and the perturbation of $f'^{(h)}(\mathbf{W})$ is small compared to the scale of $f'^{(h)}(\mathbf{W}_0)$. This property has been studied in [1] extensively. However, in the non-adversarial setting, they only need to prove this property at finitely many data points $\{\mathbf{x}_i\}_{i=1}^n$. In our adversarial training setting, though, we also need to prove that it holds for any $\mathbf{x}_i' \in \mathcal{B}(\mathbf{x}_i)$. Specifically, in this section we would like to prove that it holds for any $\mathbf{x} \in \mathcal{S}$. Our method is based on viewing the perturbation of $\mathbf{x}$ as an equivalent perturbation of the parameter $\mathbf{W}^{(1)}$, and then we will be able to make use of the results in [1]. This is elaborated in the following lemma:

**Lemma A.4.** *Given any fixed input $\mathbf{x} \in \mathcal{S}$. If $R = O(\sqrt{m})$, with probability $1 - O(H)e^{-\Omega(m/H)}$ over random initialization, for any $\mathbf{x}' \in \mathcal{S}$ satisfying $\|\mathbf{x} - \mathbf{x}'\|_2 \le \delta$, and any $\mathbf{W} \in B(R)$, there exists $\widetilde{\mathbf{W}} \in B(R + O(\sqrt{m}\delta))$ such that $\mathbf{W}^{(h)} = \widetilde{\mathbf{W}}^{(h)}$ for $h = 2, \cdots, H$, and for all $h \in [H]$ we have*

$$\overline{\mathbf{x}}'^{(h)}(\mathbf{W}) = \overline{\mathbf{x}}^{(h)}(\widetilde{\mathbf{W}}), \quad \mathbf{x}'^{(h)}(\mathbf{W}) = \mathbf{x}^{(h)}(\widetilde{\mathbf{W}}), \quad \mathbf{D}^{(h)}(\mathbf{x}', \mathbf{W}) = \mathbf{D}^{(h)}(\mathbf{x}, \widetilde{\mathbf{W}}).$$

*In other words, the network with a perturbation from $\mathbf{x}$ to $\mathbf{x}'$ is same as the network with a perturbation from $\mathbf{W}$ to $\widetilde{\mathbf{W}}$ since layer $\overline{\mathbf{x}}^{(1)}$ and up.*

*Proof.* By Lemma A.1, with probability $1 - O(H)e^{-\Omega(m)}$, $\|\mathbf{A}\|_2 = O(1)$ and $\left\|\mathbf{W}_0^{(1)}\right\|_2 = O(1)$. Thus $\left\|\mathbf{W}^{(1)}\right\|_2 \le \left\|\mathbf{W}_0^{(1)}\right\|_2 + \frac{R}{\sqrt{m}} = O(1)$. By Lemma A.2, with probability $1 - O(H)e^{-\Omega(m/H)}$, $\left\|\mathbf{x}^{(0)}\right\|_2 \in [2/3, 4/3]$. Let

$$\widetilde{\mathbf{W}}^{(1)} = \mathbf{W}^{(1)} + \frac{\mathbf{W}^{(1)}\left(\mathbf{x}'^{(0)} - \mathbf{x}^{(0)}\right)\left(\mathbf{x}^{(0)}\right)^\top}{\left\|\mathbf{x}^{(0)}\right\|_2^2}.$$

$\widetilde{\mathbf{W}}^{(1)}$ obviously satisfies $\widetilde{\mathbf{W}}^{(1)}\mathbf{x}^{(0)} = \mathbf{W}^{(1)}\mathbf{x}'^{(0)}$. Then setting $\widetilde{\mathbf{W}}^{(2)}, \ldots, \widetilde{\mathbf{W}}^{(H)}$ equal to $\mathbf{W}^{(2)}, \ldots, \mathbf{W}^{(H)}$ will make all the following hidden layer vectors and $\mathbf{D}^{(h)}$ equal. It is also easy to verify that

$$\left\|\widetilde{\mathbf{W}}^{(1)} - \mathbf{W}^{(1)}\right\|_F \le \frac{\left\|\mathbf{W}^{(1)}\right\|_2 \|\mathbf{A}\|_2 \|\mathbf{x} - \mathbf{x}'\|_2}{\left\|\mathbf{x}^{(0)}\right\|_2} = O(\delta),$$

so we know that $\widetilde{\mathbf{W}} \in B(R + O(\sqrt{m}\delta))$. $\qquad\square$

By Lemma A.4, we can directly apply many results in [1] which are only intended for the fixed data originally, to our scenario where the input can be perturbed, as long as we take the parameter radius as $\frac{R}{\sqrt{m}} + O(\delta)$ in their propositions[1]. This can give us the following important lemma:

**Lemma A.5** (Bound for the Perturbation of Gradient). *Given any fixed input $\mathbf{x} \in \mathcal{S}$. If $m \ge \max(d, \Omega(H \log H))$, $\frac{R}{\sqrt{m}} + \delta \le \frac{c}{H^6(\log m)^3}$ for some sufficiently small constant $c$, then with probability at least $1 - O(H)e^{-\Omega(m(R/\sqrt{m}+\delta)^{2/3}H)}$ over random initialization, we have for any $\mathbf{W} \in B(R)$ and any $\mathbf{x}' \in \mathcal{S}$ with $\|\mathbf{x} - \mathbf{x}'\|_2 \le \delta$,*

$$\left\|f'^{(h)}(\mathbf{W}, \mathbf{x}') - f'^{(h)}(\mathbf{W}_0, \mathbf{x})\right\|_F = O\left((\frac{R}{\sqrt{m}} + \delta)^{1/3} H^2 \sqrt{m \log m}\right)$$

*and*

$$\left\|f'^{(h)}(\mathbf{W}, \mathbf{x}')\right\|_F = O(\sqrt{mH}).$$

*Proof.* By Lemma 8.2(b)(c) of [1], using the method of Lemma A.4 stated above, when $\frac{R}{\sqrt{m}} + \delta \leq \frac{c}{H^{9/2}(\log m)^3}$, with probability $1 - e^{-\Omega(m(R/\sqrt{m}+\delta)^{2/3}H)}$, for any $\mathbf{W} \in B(R)$ and any $\mathbf{x}' \in \mathcal{S}$ with $\|\mathbf{x} - \mathbf{x}'\|_2 \leq \delta$, we have for $h \in [H]$, [2]

$$\left\|\mathbf{D}^{(h)}(\mathbf{W}, \mathbf{x}') - \mathbf{D}^{(h)}(\mathbf{W}_0, \mathbf{x})\right\|_0 = O\left(m(\frac{R}{\sqrt{m}} + \delta)^{2/3}H\right). \tag{2}$$

and

$$\left\|\mathbf{x}'^{(h)}(\mathbf{W}) - \mathbf{x}^{(h)}(\mathbf{W}_0)\right\|_2 = O\left((\frac{R}{\sqrt{m}} + \delta)H^{5/2}\sqrt{\log m}\right), \tag{3}$$

where (3) is also easily verified to hold for $h = 0$. Next, according to Lemma 8.7 of [1][3], when the bound (2) satisfies $O(m(\frac{R}{\sqrt{m}}+\delta)^{2/3}H) \leq \frac{m}{H^3 \log m}$, with probability $1 - e^{-\Omega((R/\sqrt{m}+\delta)^{2/3}Hm \log m)}$, we have for any $\mathbf{W} \in B(R)$ and any $\mathbf{x}' \in \mathcal{S}$ with $\|\mathbf{x} - \mathbf{x}'\|_2 \leq \delta, \forall h \in [H]$,

$$\left\|\mathbf{a}^\top \mathbf{D}^{(H)}(\mathbf{W}, \mathbf{x}')\mathbf{W}^{(H)} \cdots \mathbf{D}^{(h)}(\mathbf{W}, \mathbf{x}')\mathbf{W}^{(h)} - \mathbf{a}^\top \mathbf{D}^{(H)}(\mathbf{W}_0, \mathbf{x})\mathbf{W}_0^{(H)} \cdots \mathbf{D}^{(h)}(\mathbf{W}_0, \mathbf{x})\mathbf{W}_0^{(h)}\right\|_2$$
$$= O\left((\frac{R}{\sqrt{m}} + \delta)^{1/3}H^2\sqrt{m \log m}\right) \tag{4}$$

Note that with our condition $\frac{R}{\sqrt{m}} + \delta \leq \frac{c}{H^6(\log m)^3}$, the previous requirements are all satisfied. Also, combining (3) with Lemma A.2, we know for $h = 0, \cdots, H$,

$$\left\|\mathbf{x}'^{(h)}(\mathbf{W})\right\|_2 \leq O(1) + O\left((\frac{R}{\sqrt{m}} + \delta)H^{5/2}\sqrt{\log m}\right) = O(1) \tag{5}$$

Combining Equation (3), (4), (5), and Lemma A.3, we obtain

$$\left\|f'^{(h)}(\mathbf{W}, \mathbf{x}') - f'^{(h)}(\mathbf{W}_0, \mathbf{x})\right\|_F$$
$$\leq \left\|\mathbf{a}^\top \mathbf{D}^{(H)}(\mathbf{W}, \mathbf{x}')\mathbf{W}^{(H)} \cdots \mathbf{D}^{(h)}(\mathbf{W}, \mathbf{x}')\mathbf{W}^{(h)} - \mathbf{a}^\top \mathbf{D}^{(H)}(\mathbf{W}_0, \mathbf{x})\mathbf{W}_0^{(H)} \cdots \mathbf{D}^{(h)}(\mathbf{W}_0, \mathbf{x})\mathbf{W}_0^{(h)}\right\|_2$$
$$\cdot \left\|\mathbf{x}'^{(h-1)}(\mathbf{W})\right\|_2 + \left\|\mathbf{a}^\top \mathbf{D}^{(H)}(\mathbf{W}_0, \mathbf{x})\mathbf{W}_0^{(H)} \cdots \mathbf{D}^{(h)}(\mathbf{W}_0, \mathbf{x})\mathbf{W}_0^{(h)}\right\|_2 \left\|\mathbf{x}'^{(h-1)}(\mathbf{W}) - \mathbf{x}^{(h-1)}(\mathbf{W}_0)\right\|_2$$
$$= O\left((\frac{R}{\sqrt{m}} + \delta)^{1/3}H^2\sqrt{m \log m}\right) + O(\sqrt{mH}) \cdot O\left((\frac{R}{\sqrt{m}} + \delta)H^{5/2}\sqrt{\log m}\right)$$
$$= O\left((\frac{R}{\sqrt{m}} + \delta)^{1/3}H^2\sqrt{m \log m}\right).$$

In addition, also by (5) and Lemma A.3,

$$\left\|f'^{(h)}(\mathbf{W}_0, \mathbf{x})\right\|_F \leq \left\|\mathbf{a}^\top \mathbf{D}^{(H)}(\mathbf{W}_0, \mathbf{x})\mathbf{W}_0^{(H)} \cdots \mathbf{D}^{(h)}(\mathbf{W}_0, \mathbf{x})\mathbf{W}_0^{(h)}\right\|_2 \left\|\mathbf{x}^{(h-1)}(\mathbf{W}_0)\right\|_2$$
$$= O(\sqrt{mH}).$$

Therefore,

$$\left\|f'^{(h)}(\mathbf{W}, \mathbf{x}')\right\|_F = O(\sqrt{mH}) + O\left((\frac{R}{\sqrt{m}} + \delta)^{1/3}H^2\sqrt{m \log m}\right) = O(\sqrt{mH}).$$

$\square$

With Lemma A.5, we are ready to state an important bound that implies the loss function is close to being convex within the neighborhood $B(R)$ for any $\mathbf{x} \in \mathcal{S}$. We use the $\epsilon$-net to turn the result from a fixed $\mathbf{x}$ to all $\mathbf{x} \in \mathcal{S}$,

**Lemma A.6.** *If* $m = \Omega\left(\frac{d^{3/2}\log^{3/2}(\sqrt{m}/R)}{RH^{3/2}}\right)$ *and* $R = O\left(\frac{\sqrt{m}}{H^6(\log m)^3}\right)$, *then with probability at least* $1 - O(H)e^{-\Omega((mR)^{2/3}H)}$ *over random initialization, we have for any* $\mathbf{W}_1, \mathbf{W}_2 \in B(R)$, *any* $\mathbf{x} \in \mathcal{S}$ *and any* $y \in \mathbb{R}$,

$$l\left(f(\mathbf{W}_2, \mathbf{x}), y\right) \geq l\left(f(\mathbf{W}_1, \mathbf{x}), y\right) + \langle \nabla_{\mathbf{W}} l(f(\mathbf{W}_1, \mathbf{x}), y), \mathbf{W}_2 - \mathbf{W}_1 \rangle$$
$$- \|\mathbf{W}_2 - \mathbf{W}_1\|_F \, O((mR)^{1/3} H^{5/2} \sqrt{\log m}).$$

*Proof.* A $\delta$-net of $\mathcal{S}$ is a group of points $\{\mathbf{x}_i\}_{i=1}^N$ in $\mathcal{S}$ that satisfies: for any $\mathbf{x} \in \mathcal{S}$, there exists some $\mathbf{x}_i$ that satisfies $\|\mathbf{x} - \mathbf{x}_i\|_2 \leq \delta$. From classic covering number results we know that we can construct such a $\delta$-net with the number of total points $N = (O(1/\delta))^d$, where $d$ is the input dimension. As long as $\frac{R}{\sqrt{m}} + \delta \leq \frac{c}{H^6(\log m)^3}$ for some sufficiently small constant $c$, we can derive that for any $i \in [N]$, with probability $1 - O(H)e^{-\Omega(m(R/\sqrt{m}+\delta)^{2/3}H)}$, for any $\mathbf{W}_1, \mathbf{W}_2 \in B(R)$, any $\mathbf{x}_i' \in \mathcal{S}$ with $\|\mathbf{x}_i - \mathbf{x}_i'\|_2 \leq \delta$, and any $y \in \mathbb{R}$,

$$l\left(f(\mathbf{W}_2, \mathbf{x}_i'), y\right) - l\left(f(\mathbf{W}_1, \mathbf{x}_i'), y\right) - \langle \nabla_{\mathbf{W}} l(f(\mathbf{W}_1, \mathbf{x}_i'), y), \mathbf{W}_2 - \mathbf{W}_1 \rangle$$
$$\geq \frac{\partial}{\partial f} l\left(f(\mathbf{W}_1, \mathbf{x}_i'), y\right) \left[f(\mathbf{W}_2, \mathbf{x}_i') - f(\mathbf{W}_1, \mathbf{x}_i') - \langle \nabla_{\mathbf{W}} f(\mathbf{W}_1, \mathbf{x}_i'), \mathbf{W}_2 - \mathbf{W}_1 \rangle\right]$$
$$= \frac{\partial}{\partial f} l\left(f(\mathbf{W}_1, \mathbf{x}_i'), y\right) \langle \int_0^1 (\nabla_{\mathbf{W}} f(t\mathbf{W}_2 + (1-t)\mathbf{W}_1, \mathbf{x}_i') - \nabla_{\mathbf{W}} f(\mathbf{W}_1, \mathbf{x}_i')) \, dt, \mathbf{W}_2 - \mathbf{W}_1 \rangle$$
$$\geq - \|\mathbf{W}_2 - \mathbf{W}_1\|_F \, O((\frac{R}{\sqrt{m}} + \delta)^{1/3} H^{5/2} \sqrt{m \log m}),$$

where the first inequality uses the convexity of $l$ w.r.t $f$ and the last inequality is due to the boundedness of $|\partial l / \partial f|$ is bounded, Lemma A.5, and $\nabla_{\mathbf{W}} f = (f'^{(1)}, \dots, f'^{(H)})$. We take $\delta = \frac{R}{\sqrt{m}}$. With $R = O\left(\frac{\sqrt{m}}{H^6(\log m)^3}\right)$ the requirement $\frac{R}{\sqrt{m}} + \delta \leq \frac{c}{H^6(\log m)^3}$ can be satisfied. Therefore, taking union event over all $N$ points, our proposition holds with probability

$$1 - O(H)(O(\sqrt{m}/R))^d e^{-\Omega((mR)^{2/3}H)}$$
$$= 1 - O(H)e^{-\Omega((mR)^{2/3}H) + d\log(O(\sqrt{m}/R))}$$
$$= 1 - O(H)e^{-\Omega((mR)^{2/3}H)},$$

where the last equation is due to the condition $m = \Omega\left(\frac{d^{3/2}\log^{3/2}(\sqrt{m}/R)}{RH^{3/2}}\right)$. $\qquad\square$

With the above preparations, we are ready to prove the main theorem.

*Proof of Theorem 4.1.* We denote $\mathbf{W}_t$ as the parameter after $t$ steps of projected gradient descent, starting from the initialization $\mathbf{W}_0$. We perform a total of $T$ steps with step size $\alpha$.

For projected gradient descent, $\mathbf{W}_t \in B(R)$ holds for all $t = 0, 1, \cdots, T$. Recall that the update rule is $\mathbf{W}_{t+1} = \mathcal{P}_{B(R)}(\mathbf{V}_{t+1})$ for $\mathbf{V}_{t+1} = \mathbf{W}_t - \alpha \nabla_{\mathbf{W}} L_{\mathcal{A}}(\mathbf{W}_t)$. Let $d_t := \|\mathbf{W}_t - \mathbf{W}_*\|_F$. We have

$$
\begin{aligned}
d_{t+1}^2 =& \|\mathbf{W}_{t+1} - \mathbf{W}_*\|_F^2 \\
\leq& \|\mathbf{V}_{t+1} - \mathbf{W}_*\|_F^2 \\
=& \|\mathbf{W}_t - \mathbf{W}_*\|_F^2 + 2\langle \mathbf{V}_{t+1} - \mathbf{W}_t, \mathbf{W}_t - \mathbf{W}_* \rangle + \|\mathbf{V}_{t+1} - \mathbf{W}_t\|_F^2 \\
=& d_t^2 + 2\alpha \langle \nabla_{\mathbf{W}} L_{\mathcal{A}}(\mathbf{W}_t), (\mathbf{W}_* - \mathbf{W}_t) \rangle + \alpha^2 \|\nabla_{\mathbf{W}} L_{\mathcal{A}}(\mathbf{W}_t)\|_F^2 \\
=& d_t^2 + \frac{2\alpha}{n} \sum_{i=1}^n \langle \nabla_{\mathbf{W}} l(\mathbf{W}_t, \mathcal{A}(\mathbf{W}_t, \mathbf{x}_i)), (\mathbf{W}_* - \mathbf{W}_t) \rangle + \alpha^2 \| \frac{1}{n} \sum_{i=1}^n \frac{\partial l}{\partial f} \nabla_{\mathbf{W}} f(\mathbf{W}_t, \mathcal{A}(\mathbf{W}_t, \mathbf{x}_i)) \|_F^2 \\
\leq& d_t^2 + \frac{2\alpha}{n} \sum_{i=1}^n [l(\mathbf{W}_*, \mathcal{A}(\mathbf{W}_t, \mathbf{x}_i)) - l(\mathbf{W}_t, \mathcal{A}(\mathbf{W}_t, \mathbf{x}_i)) \\
& + \|\mathbf{W}_* - \mathbf{W}_t\|_F \, O((mR)^{1/3} H^{5/2} \sqrt{\log m})] + \alpha^2 O(mH^2) \\
\leq& d_t^2 + \frac{2\alpha}{n} \sum_{i=1}^n (l(\mathbf{W}_*, \mathcal{A}_*(\mathbf{W}_*, \mathbf{x}_i)) - l(\mathbf{W}_t, \mathcal{A}(\mathbf{W}_t, \mathbf{x}_i))) \\
& + O(\alpha m^{-1/6} R^{4/3} H^{5/2} \sqrt{\log m} + \alpha^2 m H^2) \\
=& d_t^2 + 2\alpha (L_*(\mathbf{W}_*) - L_{\mathcal{A}}(\mathbf{W}_t)) + O(\alpha m^{-1/6} R^{4/3} H^{5/2} \sqrt{\log m} + \alpha^2 m H^2)
\end{aligned}
$$

where the second inequality is due to Lemma A.6 and Lemma A.5, the third inequality is due to the definition of $\mathcal{A}_*$. Note that in order to satisfy the condition for Lemma A.6 and Lemma A.5, our choice $m = \max\{\Omega\left(\frac{H^{16} R^9}{\epsilon^7}\right), \Omega(d^2)\}$ suffices. By induction on the above inequality, we have

$$
d_T^2 \leq d_0^2 + 2\alpha \sum_{t=0}^{T-1} (L_*(\mathbf{W}_*) - L_{\mathcal{A}}(\mathbf{W}_t)) + O(T(\alpha m^{-1/6} R^{4/3} H^{5/2} \sqrt{\log m} + \alpha^2 m H^2)),
$$

which implies that

$$
\begin{aligned}
\min_{0 \leq t \leq T} (L_*(\mathbf{W}_*) - L_{\mathcal{A}}(\mathbf{W}_t)) &\leq \frac{d_0^2 - d_T^2}{\alpha T} + O(m^{-1/6} R^{4/3} H^{5/2} \sqrt{\log m} + \alpha m H^2) \\
&\leq \frac{R^2}{m \alpha T} + O(m^{-1/6} R^{4/3} H^{5/2} \sqrt{\log m} + \alpha m H^2) \\
&\leq \epsilon,
\end{aligned}
$$

where in the last inequality we use our choice of $\alpha = O\left(\frac{\epsilon}{mH^2}\right)$, $T = \Theta\left(\frac{R^2}{m\alpha\epsilon}\right)$, and also $m^{-1/6} R^{4/3} H^{5/2} \sqrt{\log m} \leq O(\epsilon)$, which is satisfied by $m = \Omega\left(\frac{H^{16} R^9}{\epsilon^7}\right)$.

## B  Proof of Theorem 5.1: Convergence Result for Two-Layer Networks

*Proof.* We denote $\mathbf{W}_t$ as the parameter after $t$ steps of projected gradient descent, starting from the initialization $\mathbf{W}_0$. We perform a total of $T$ steps with step size $\alpha$, where each step is an update $\mathbf{W}_{t+1} = \mathbf{W}_t - \alpha \nabla_{\mathbf{W}} L_{\mathcal{A}}(\mathbf{W}_t)$. Firstly, the formula for the network gradient is

$$
\nabla_{\mathbf{W}} f(\mathbf{W}, \mathbf{x}) = \frac{1}{\sqrt{m}} \operatorname{diag}(\mathbf{a}) \sigma'(\mathbf{W}\mathbf{x}) \mathbf{x}^\top,
$$

where $\mathbf{a} = (a_1, \cdots, a_{\frac{m}{2}}, a'_1, \cdots, a'_{\frac{m}{2}})^\top$ is the parameter for the output layer. We can compute the Lipschitz property of $\nabla_{\mathbf{W}} f$ w.r.t $\mathbf{W}$: For any fixed $\mathbf{x} \in \mathcal{S}$,

$$
\begin{aligned}
\|\nabla_{\mathbf{W}} f(\mathbf{W}) - \nabla_{\mathbf{W}} f(\mathbf{W}')\|_F &\leq \frac{1}{\sqrt{m}} \|\operatorname{diag}(\mathbf{a})\|_2 \|\sigma'(\mathbf{W}\mathbf{x})) - \sigma'(\mathbf{W}'\mathbf{x})\|_2 \|\mathbf{x}\|_2 \\
&\leq \frac{1}{\sqrt{m}} \cdot 1 \cdot C \|\mathbf{W}\mathbf{x} - \mathbf{W}'\mathbf{x}\|_2 \cdot 1 \\
&\leq O\left(\frac{1}{\sqrt{m}}\right) \|\mathbf{W} - \mathbf{W}'\|_F,
\end{aligned}
$$

For a fixed data point $(\mathbf{x}, y)$, we denote $\ell(f(\mathbf{W}))$ as short for $\ell(f(\mathbf{W}, \mathbf{x}), y)$. Since $\ell$ is convex and has bounded derivative in $f$, we have

$$
\begin{aligned}
&\ell(f(\mathbf{W}')) - \ell(f(\mathbf{W})) \\
\geq & \ell'(f(\mathbf{W}))(f(\mathbf{W}') - f(\mathbf{W})) \\
= & \ell'(f(\mathbf{W}))(\langle \nabla_{\mathbf{W}} f(\mathbf{W}), \mathbf{W}' - \mathbf{W} \rangle + \langle \int_0^1 (\nabla_{\mathbf{W}} f(s\mathbf{W} + (1-s)\mathbf{W}') - \nabla_{\mathbf{W}} f(\mathbf{W}))ds, \mathbf{W}' - \mathbf{W} \rangle) \\
\geq & \langle \nabla_{\mathbf{W}} \ell(f(\mathbf{W})), \mathbf{W} - \mathbf{W}' \rangle - O\left(\frac{1}{\sqrt{m}}\right) \|\mathbf{W} - \mathbf{W}'\|_F^2 .
\end{aligned}
\tag{6}
$$

In addition, we can also easily know that for $\mathbf{W} \in B(R)$ and $R = O(\sqrt{M})$, $\mathbf{x} \in \mathcal{S}$, we have

$$
\|\nabla_{\mathbf{W}} \ell(f(\mathbf{W}))\|_2 \leq |\ell'| \frac{1}{\sqrt{m}} \|\text{diag}(\mathbf{a})\|_2 \left(\sqrt{m}|\sigma'(0)| + C \|\mathbf{W}\mathbf{x}\|_2\right) \|\mathbf{x}\|_2 = O(1)
\tag{7}
$$

since the initialization satisfies $\|\mathbf{W}_0\|_2 = O(\sqrt{m})$ with high probability given $m = \Omega(d)$ (see Lemma A.1), thereby $\|\mathbf{W}\|_2 = O(\sqrt{m})$.

Denote $d_t = \|\mathbf{W}_t - \mathbf{W}_*\|_F$. Without a projection step, there could be two possible scenarios during the optimization process: Either $\mathbf{W}_t \in B(3R)$ holds for all $t = 1, \cdots, T$, or there exists some $T_0 < T$ such that $\mathbf{W}_t \in B(3R)$ for $t \leq T_0$ but $\mathbf{W}_{T_0+1} \notin B(3R)$. Either way, while $\mathbf{W}_t$ is still in $B(3R)$ up to $t - 1$, we have

$$
\begin{aligned}
d_t^2 &= \|\mathbf{W}_t - \mathbf{W}_*\|_F^2 \\
&= \|\mathbf{W}_{t-1} - \mathbf{W}_*\|_F^2 + 2(\mathbf{W}_t - \mathbf{W}_{t-1}) \cdot (\mathbf{W}_{t-1} - \mathbf{W}_*) + \|\mathbf{W}_t - \mathbf{W}_{t-1}\|_F^2 \\
&= d_{t-1}^2 + 2\alpha \nabla_{\mathbf{W}} L_{\mathcal{A}}(\mathbf{W}_{t-1}) \cdot (\mathbf{W}_* - \mathbf{W}_{t-1}) + \alpha^2 \|\nabla_{\mathbf{W}} L_{\mathcal{A}}(\mathbf{W}_{t-1})\|_F^2 \\
&\leq d_{t-1}^2 + \frac{2\alpha}{n} \sum_{i=1}^n \left[ \ell(f(\mathbf{W}_*, \mathcal{A}(\mathbf{W}_{t-1}, \mathbf{x}_i)), y_i) - \ell(f(\mathbf{W}_{t-1}, \mathcal{A}(\mathbf{W}_{t-1}, \mathbf{x}_i)), y_i) + O\left(\frac{1}{\sqrt{m}}\right) \|\mathbf{W}_* - \mathbf{W}_{t-1}\|_F^2 \right] + O(\alpha \\
&\leq \left(1 + \frac{c\alpha}{\sqrt{m}}\right) d_{t-1}^2 + 2\alpha(L_*(\mathbf{W}_*) - L_{\mathcal{A}}(\mathbf{W}_{t-1})) + O(\alpha^2),
\end{aligned}
\tag{8}
$$

where the first inequality is based on (6) and (7), the second inequality is based on the definition of $L_*(\mathbf{W}_*)$, and $c$ is some constant. Let $S_t = (1 + \frac{c\alpha}{\sqrt{m}})^t$ which is a geometric series, and dividing (8) by $S_t$ we have

$$
\frac{d_t^2}{S_t} \leq \frac{d_{t-1}^2}{S_{t-1}} - 2\alpha \frac{L_{\mathcal{A}}(\mathbf{W}_{t-1}) - L_*(\mathbf{W}_*)}{S_t} + \frac{O(\alpha^2)}{S_t},
$$

which, by induction, gives us

$$
\begin{aligned}
\frac{d_t^2}{S_t} &\leq d_0^2 - 2\alpha \sum_{i=0}^{t-1} \frac{L_{\mathcal{A}}(\mathbf{W}_i) - L_*(\mathbf{W}_*)}{S_{i+1}} + O(\alpha^2) \sum_{i=0}^{t-1} \frac{1}{S_{i+1}} \\
&\leq d_0^2 - 2\alpha \min_{i=0,\cdots,t-1} (L_{\mathcal{A}}(\mathbf{W}_i) - L_*(\mathbf{W}_*)) \sum_{i=0}^{t-1} \frac{1}{S_{i+1}} + O(\alpha^2) \sum_{i=0}^{t-1} \frac{1}{S_{i+1}},
\end{aligned}
$$

and note that $\sum_{i=0}^{t-1} \frac{1}{S_{i+1}} = \frac{\sqrt{m}}{c\alpha} \left(1 - \frac{1}{S_t}\right)$, which yields

$$
\min_{i=0,\cdots,t-1} L_{\mathcal{A}}(\mathbf{W}_i) - L_*(\mathbf{W}_*) \leq O(\alpha) + \frac{c\left(d_0^2 - \frac{d_t^2}{S_t}\right)}{\sqrt{m}(1 - \frac{1}{S_t})}.
\tag{9}
$$

Now we will consider the two cases separately:

*Case 1.* $\mathbf{W}_t \in B(3R)$ holds for all $t = 1, \cdots, T$. We have chosen $T = \frac{\sqrt{m}}{c\alpha}$, and then $S_T \approx e$. Also, since $d_0^2 - \frac{d_T^2}{S_T} \leq d_0^2 = O(R^2)$, by choosing $m = \Omega(R^4/\epsilon^2)$ and $\alpha = O(\epsilon)$, and taking $t = T$ in (9), we can obtain $\min_{t=0,\cdots,T} L_{\mathcal{A}}(\mathbf{W}_t) - L_*(\mathbf{W}_*) \leq \epsilon$.

*Case 2.* There exists some $T_0 < T$ such that $\mathbf{W}_t \in B(3R)$ for $t \leq T_0$ but $\mathbf{W}_{T_0+1} \notin B(3R)$. Since $\mathbf{W}_* \in B(R)$, we know that $d_0 \leq R$ and $d_{T_0+1} \geq 2R$. Still using the choice of parameters above, we have $d_0^2 - \frac{d_{T_0+1}^2}{S_{T_0+1}} \leq R^2 - (4/e)R^2 \leq 0$. Hence, taking $t = T_0 + 1$ in (9), we obtain $\min_{t=0,\cdots,T} L_{\mathcal{A}}(\mathbf{W}_t) - L_*(\mathbf{W}_*) \leq \min_{t=0,\cdots,T_0} L_{\mathcal{A}}(\mathbf{W}_t) - L_*(\mathbf{W}_*) \leq \epsilon$.

So in any case the result is correct, thus we have proved the convergence without the need of projection. $\qquad\square$

## C  Proof of Gradient Descent Finding Robust Classifier in Section 5

### C.1  Proof of Theorem 5.2

As discussed in Section 5, we will use the idea of random feature [7] to approximate $g \in \mathcal{H}(K_\sigma)$ on the unit sphere. We consider functions of the form

$$h(\mathbf{x}) = \int_{\mathbb{R}^d} c(\mathbf{w})^\top \mathbf{x} \sigma'(\mathbf{w}^\top \mathbf{x}) d\mathbf{w},$$

where $c(\mathbf{w}) : \mathbb{R}^d \to \mathbb{R}^d$ is any function from $\mathbb{R}^d$ to $\mathbb{R}^d$. We define the RF-norm of $h$ as $\|h\|_{\mathrm{RF}} = \sup_{\mathbf{w}} \frac{\|c(\mathbf{w})\|_2}{p_0(\mathbf{w})}$ where $p_0(\mathbf{w})$ is the probability density function of $\mathcal{N}(0, \mathbf{I}_d)$, which is the distribution of initialization. Define the function class with finite $\mathcal{N}(0, \mathbf{I}_d)$-norm as $\mathcal{F}_{\mathrm{RF}} = \left\{ h(\mathbf{x}) = \int_{\mathbb{R}^d} c(\mathbf{w})^\top \mathbf{x} \sigma'(\mathbf{w}^\top \mathbf{x}) d\mathbf{w} : \|h\|_{\mathrm{RF}} < \infty \right\}$. We firstly show that $\mathcal{F}_{\mathrm{RF}}$ is dense in $\mathcal{H}(K_\sigma)$.

**Lemma C.1** (Universality of $\mathcal{F}_{\mathrm{RF}}$). *Let $\mathcal{F}_{\mathrm{RF}}$ and $\mathcal{H}(K_\sigma)$ be defined as above. Then $\mathcal{F}_{\mathrm{RF}}$ is dense in $\mathcal{H}(K_\sigma)$, and further, dense in $\mathcal{H}(K_\sigma)$ w.r.t. $\|\cdot\|_{\infty,\mathcal{S}}$, where $\|f\|_{\infty,\mathcal{S}} = \sup_{\mathbf{x}\in\mathcal{S}} |f(\mathbf{x})|$.*

*Proof.* Observe that by the definition of the RKHS introduced by $K_\sigma$, functions with form $h(\mathbf{x}) = \sum_t a_t K(\mathbf{x}, \mathbf{x}_t)$, $\mathbf{x}_t \in \mathcal{S}$ are dense in $\mathcal{H}(K_\sigma)$. But these functions can also be written in the form $h(\mathbf{x}) = \int_{\mathbb{R}^d} c(\mathbf{w})^\top \mathbf{x} \sigma'(\mathbf{w}^\top \mathbf{x}) d\mathbf{w}$ where $c(\mathbf{w}) = p_0(\mathbf{w}) \sum_t a_t \mathbf{x}_t \sigma'(\mathbf{w}^\top \mathbf{x}_t)$. Note that $\|c(\mathbf{w})\|_2 \leq p(\mathbf{w}) \sum_t \left\| a_t \mathbf{x}_t \sigma'(\mathbf{w}^\top \mathbf{x}_t) \right\|_2 < \infty$ since $\mathcal{S}$ is a compact set and $\sigma'$ is bounded, this verifies that $h$ is an element in $\mathcal{F}_{\mathrm{RF}}$. So $\mathcal{F}_{\mathrm{RF}}$ contains a dense set of $\mathcal{H}(K_\sigma)$ and therefore dense in $\mathcal{H}(K_\sigma)$. Then note that the evaluation operator $K_{\sigma,\mathbf{x}}$ is uniformly bounded for $\mathbf{x} \in \mathcal{S}$, and $h(\mathbf{x}) = \langle K_{\sigma,\mathbf{x}}, h \rangle_{\mathcal{H}}$, so the RKHS norm can be used to control the norm $\|\cdot\|_{\infty,\mathcal{S}}$ and is therefore stronger, thus the proof is complete. $\qquad\square$

We then show that we can approximate elements of $\mathcal{F}_{\mathrm{RF}}$ by finite random features. Our results are inspired by [7]. For the next theorem, recall Assumption 5.1, 5.3, the constant $C$ satisfies $\sigma'$ is $C$-Lipschitz, $|\sigma'(\cdot)| \leq C$.

**Proposition C.1** (Approximation by Finite Sum). *Let $h(\mathbf{x}) = \int_{\mathbb{R}^d} c(\mathbf{w})^\top \mathbf{x} \sigma'(\mathbf{w}^\top \mathbf{x}) d\mathbf{w} \in \mathcal{F}_{RF}$. Then for any $\delta > 0$, with probability at least $1 - \delta$ over $\mathbf{w}_1, \cdots, \mathbf{w}_M$ drawn i.i.d. from $\mathcal{N}(0, \mathbf{I}_d)$, there exists $c_1, \cdots, c_M$ where $c_i \in \mathbb{R}^d$ and $\|c_i\|_2 \leq \frac{\|h\|_{RF}}{M}$, so that the function $\hat{h} = \sum_{i=1}^M c_i^\top \mathbf{x} \sigma'(\mathbf{w}_i^\top \mathbf{x})$, satisfies*

$$\left\| \hat{h} - h \right\|_{\infty,\mathcal{S}} \leq \frac{C \|h\|_{RF}}{\sqrt{M}} \left( 2\sqrt{d} + \sqrt{2\log(1/\delta)} \right).$$

*Proof.* This result is obtained by importance sampling, where we construct $\hat{h}$ with $c_i = \frac{c(\mathbf{w}_i)}{M p_0(\mathbf{w}_i)}$. We first notice that $\|c_i\|_2 = \frac{\|c(\mathbf{w}_i)\|_2}{M p_0(\mathbf{w}_i)} \leq \frac{\|h\|_{\mathrm{RF}}}{M}$ which satisfies the condition of the theorem. We then define the random variable

$$v(\mathbf{w}_1, \cdots, \mathbf{w}_M) = \left\| \hat{h} - h \right\|_{\infty,\mathcal{S}}.$$

We bound this deviation from its expectation using McDiarmid's inequality.

To do so, we should first show that $v$ is robust to the perturbation of one of its arguments. In fact, for $\mathbf{w}_1, \cdots, \mathbf{w}_M$ and $\tilde{\mathbf{w}}_i$ we have

$$|v(\mathbf{w}_1, \cdots, \mathbf{w}_M) - v(\mathbf{w}_1, \cdots, \tilde{\mathbf{w}}_i, \cdots, \mathbf{w}_M)|$$

$$\leq \frac{1}{M} \max_{\mathbf{x} \in \mathcal{S}} \left| \frac{c(\mathbf{w}_i)^\top \mathbf{x} \sigma'(\mathbf{w}_i^\top \mathbf{x})}{p_0(\mathbf{w}_i)} - \frac{c(\tilde{\mathbf{w}}_i)^\top \mathbf{x} \sigma'(\tilde{\mathbf{w}}_i^\top \mathbf{x})}{p_0(\tilde{\mathbf{w}}_i)} \right|$$

$$\leq \frac{1}{M} \|h\|_{\mathrm{RF}} \max_{\mathbf{x} \in \mathcal{S}} \left( \left| \sigma'(\mathbf{w}_i^\top \mathbf{x}) \right| + \left| \sigma'(\tilde{\mathbf{w}}_i^\top \mathbf{x}) \right| \right)$$

$$\leq \frac{2C \|h\|_{\mathrm{RF}}}{M} =: \xi$$

by using triangle, Cauchy-Schwartz inequality, $|\sigma'(\cdot)| \leq C$ and $\|\mathbf{x}\|_2 = 1$.

Next, we bound the expectation of $v$. First, observe that the choice of $c_1, \cdots, c_M$ ensures that $\mathbb{E}_{\mathbf{w}_1, \cdots, \mathbf{w}_M} \hat{h} = h$. By symmetrization [5], we have

$$\mathbb{E}v = \mathbb{E} \sup_{\mathbf{x} \in \mathcal{S}} \left| \hat{h}(\mathbf{x}) - \mathbb{E}\hat{h}(\mathbf{x}) \right|$$

$$\leq 2\mathbb{E}_{\mathbf{w}, \epsilon} \sup_{\mathbf{x} \in \mathcal{S}} \left| \sum_{i=1}^{M} \epsilon_i c_i^\top \mathbf{x} \sigma'(\mathbf{w}_i^\top \mathbf{x}) \right|, \tag{10}$$

where $\epsilon_1, \cdots, \epsilon_M$ is a sequence of Rademacher random variables.

Since $\left| c_i^\top \mathbf{x} \right| \leq \|c_i\|_2 \leq \frac{\|h\|_{\mathrm{RF}}}{M}$ and $\sigma'$ is $C$-Lipschitz, we have that $c_i^\top \mathbf{x} \sigma'(\cdot)$ is $\frac{C\|h\|_{\mathrm{RF}}}{M}$-Lipschitz in the scalar argument and zero when the scalar argument is zero. Following (10), by Talagrand's lemma (Lemma 5.7) in [6] together with Cauchy-Schwartz, Jensen's inequality, we have

$$\mathbb{E}v \leq 2\mathbb{E}_{\mathbf{w}, \epsilon} \sup_{\mathbf{x} \in \mathcal{S}} \left| \sum_{i=1}^{M} \epsilon_i c_i^\top \mathbf{x} \sigma'(\mathbf{w}_i^\top \mathbf{x}) \right|$$

$$\leq \frac{2C \|h\|_{\mathrm{RF}}}{M} \mathbb{E} \sup_{\mathbf{x} \in \mathcal{S}} \left| \sum_{i=1}^{M} \epsilon_i \mathbf{w}_i^\top \mathbf{x} \right|$$

$$\leq \frac{2C \|h\|_{\mathrm{RF}}}{M} \mathbb{E} \left\| \sum_{i=1}^{M} \epsilon_i \mathbf{w}_i \right\|_2$$

$$\leq \frac{2C \|h\|_{\mathrm{RF}}}{\sqrt{M}} \sqrt{\mathbb{E}_{\mathbf{w} \sim \mathcal{N}(0, \mathbf{I}_d)} \|\mathbf{w}\|_2^2} =: \mu$$

Then McDiarmid's inequality implies

$$\mathbb{P}[v \geq \mu + \epsilon] \leq \mathbb{P}[v \geq \mathbb{E}v + \epsilon] \leq \exp(-\frac{2\epsilon^2}{M\xi}).$$

The proposition is proved by solving the $\epsilon$ while setting the right hand to the given $\delta$. $\qquad \square$

**Proof of Theorem 5.2** . Finally, we construct $\mathbf{W}_*$ within a ball of the initialization $\mathbf{W}_0$ that suffers little robust loss $L_*(\mathbf{W}_*)$. Using the symmetric initialization in (4), we have $f(\mathbf{W}_0, \mathbf{x}) = 0$ for all $\mathbf{x}$. We then use the neural Taylor expansion w.r.t. the parameters:

$$f(\mathbf{W}, \mathbf{x}) - f(\mathbf{W}_0, \mathbf{x}) \approx \frac{1}{\sqrt{m}} \underbrace{\left( \sum_{i=1}^{m/2} a_i (\mathbf{w}_i - \mathbf{w}_{i0})^\top \mathbf{x} \sigma'(\mathbf{w}_{i0}^\top \mathbf{x}) + \sum_{i=1}^{m/2} a_i' (\bar{\mathbf{w}}_i - \bar{\mathbf{w}}_{i0})^\top \mathbf{x} \sigma'(\bar{\mathbf{w}}_{i0}^\top \mathbf{x}) \right)}_{(i)},$$

where $\mathbf{w}_{i0}$ denotes the value of $\mathbf{w}_i$ at initialization. We omitted the second order term. The term (i) has the form of the random feature approximation, and so Proposition C.1 can be used to construct a robust interpolant.

In summary, we give the entire proof of Theorem 5.2 as follows.

*Proof.* Let $L$ be the Lipschitz coefficient of the loss function $\ell$. Let $\bar{\epsilon} = \frac{1}{3L}$.

By Assumption 5.2 with $\bar{\epsilon}$, there exists $g_1 \in \mathcal{H}(K_\sigma)$ such that

$$|g_1(\mathbf{x}_i') - y_i| \leq \bar{\epsilon},$$

for every $\mathbf{x}_i' \in \mathcal{B}(\mathbf{x}_i)$, $i \in [n]$, where $\mathcal{B}(\mathbf{x}_i)$ is the perturbation set.

By Lemma C.1, for $\bar{\epsilon}$ there exists $g_2 \in \mathcal{F}_{\mathrm{RF}}$ such that $\|g_1 - g_2\|_{\infty,\mathcal{S}} \leq \bar{\epsilon}$. Then, by Theorem C.1, we have $c_1, \cdots, c_{m/2}$ where $c_i \in \mathbb{R}^d$ and

$$\|c_i\|_2 \leq \frac{\|g_2\|_{\mathrm{RF}}}{m},$$

such that $g_3 = \sum_{i=1}^{m/2} c_i^\top \mathbf{x} \sigma'(\mathbf{w}_i^\top \mathbf{x})$ satisfies

$$\|g_2 - g_3\|_{\infty,\mathcal{S}} \leq \frac{C\,\|g_2\|_{\mathrm{RF}}}{\sqrt{m/2}} \left( 2\sqrt{d} + \sqrt{2\log 1/\delta} \right),$$

with probability at least $1 - \delta$ on the initialization $\mathbf{w}_i$'s.

We decompose $f$ into the linear part and its residual:

$$
\begin{aligned}
f(\mathbf{W}, \mathbf{x}) = &\frac{1}{\sqrt{m}} \left( \sum_{i=1}^{m/2} a_i (\mathbf{w}_i - \mathbf{w}_{i0})^\top \mathbf{x} \sigma'(\mathbf{w}_{i0}^\top \mathbf{x}) + \sum_{i=1}^{m/2} a_i'(\bar{\mathbf{w}}_i - \bar{\mathbf{w}}_{i0})^\top \mathbf{x} \sigma'(\bar{\mathbf{w}}_{i0}^\top \mathbf{x}) \right) \\
&+ \frac{1}{\sqrt{m}} \Big( \sum_{i=1}^{m/2} a_i \int_0^1 \mathbf{x} \left( \sigma'((t\mathbf{w}_i + (1-t)\mathbf{w}_{i0})^\top \mathbf{x}) - \sigma'(\mathbf{w}_{i0}^\top \mathbf{x}) \right) dt \\
&+ \sum_{i=1}^{m/2} a_i' \int_0^1 \mathbf{x} \left( \sigma'((t\bar{\mathbf{w}}_i + (1-t)\bar{\mathbf{w}}_{i0})^\top \mathbf{x}) - \sigma'(\bar{\mathbf{w}}_{i0}^\top \mathbf{x}) \right) dt \Big),
\end{aligned}
$$

Then set $\mathbf{w}_i = \mathbf{w}_{i0} + \sqrt{\frac{m}{4}} c_i$, $\bar{\mathbf{w}}_i = -\sqrt{\frac{m}{4}} c_i + \bar{\mathbf{w}}_{i0}$, we have

$$\|\mathbf{w}_r - \mathbf{w}_{r0}\|_2 \leq \frac{\|g_2\|_{\mathrm{RF}}}{\sqrt{4m}},$$

$$
\begin{aligned}
\text{and } &\frac{1}{\sqrt{m}} \left( \sum_{i=1}^{m/2} a_i (\mathbf{w}_i - \mathbf{w}_{i0})^\top \mathbf{x} \sigma'(\mathbf{w}_{i0}^\top \mathbf{x}) + \sum_{i=1}^{m/2} a_i'(\bar{\mathbf{w}}_i - \bar{\mathbf{w}}_{i0})^\top \mathbf{x} \sigma'(\bar{\mathbf{w}}_{i0}^\top \mathbf{x}) \right) \\
&= \frac{1}{\sqrt{m}} \left( \sum_{i=1}^{m/2} a_i \sqrt{\frac{m}{4}} c_i^\top \mathbf{x} \sigma'(\mathbf{w}_{i0}^\top \mathbf{x}) - \sum_{i=1}^{m/2} a_i' \sqrt{\frac{m}{4}} c_i^\top \mathbf{x} \sigma'(\bar{\mathbf{w}}_{i0}^\top \mathbf{x}) \right) \\
&= \sum_{i=1}^{m} c_i^\top \mathbf{x} \sigma'(\mathbf{w}_i^\top \mathbf{x}) \\
&= g_3,
\end{aligned}
$$

So

$$
\begin{aligned}
\|f(\mathbf{W}, x) - g_3\|_{\infty,\mathcal{S}} = &\Big\| \frac{1}{\sqrt{m}} \Big( \sum_{i=1}^{m/2} a_i \int_0^1 \mathbf{x} \left( \sigma'((t\mathbf{w}_i + (1-t)\mathbf{w}_{i0})^\top \mathbf{x}) - \sigma'(\mathbf{w}_{i0}^\top \mathbf{x}) \right) dt \\
&+ \sum_{i=1}^{m/2} a_i' \int_0^1 \mathbf{x} \left( \sigma'((t\bar{\mathbf{w}}_i + (1-t)\bar{\mathbf{w}}_{i0})^\top \mathbf{x}) - \sigma'(\bar{\mathbf{w}}_{i0}^\top \mathbf{x}) \right) dt \Big) \Big\|_{\infty,\mathcal{S}} \\
\leq &\frac{1}{\sqrt{m}} \left\| m \times C \left| (t\mathbf{w}_i + (1-t)\mathbf{w}_{i0})^\top \mathbf{x} - \mathbf{w}_{i0}^\top \mathbf{x} \right| \|\mathbf{x}\| \|\mathbf{w}_i - \mathbf{w}_{i0}\| \right\|_{\infty,\mathcal{S}} \\
\leq &\frac{C\,\|g_2\|_{\mathrm{RF}}^2}{4\sqrt{m}},
\end{aligned}
$$

and therefore

$$\|f(\mathbf{W}, x) - g_2\|_{\infty, \mathcal{S}} \leq \frac{C \|g_2\|_{\mathrm{RF}}^2}{4\sqrt{m}} + \frac{C \|g_2\|_{\mathrm{RF}}}{\sqrt{m/2}} \left( 2\sqrt{d} + \sqrt{2\log(1/\delta)} \right). \tag{11}$$

Finally, set $m$ to be large enough ($= \Omega \left( \frac{\|g_2\|_{\mathrm{RF}}^4}{\epsilon^2} \right)$) so that the left hand in Equation (11) no more than $\bar{\epsilon}$ and let $R_{\mathcal{D}, \mathcal{B}, \epsilon}$ to be $\|g_2\|_{\mathrm{RF}} / 2$. Then

$$
\begin{aligned}
L_*(\mathbf{W}) &= \frac{1}{n} \sum_{i=1}^{n} \sup_{\mathbf{x} \in \mathcal{B}(\mathbf{x}_i)} \ell\left(f(\mathbf{W}, \mathbf{x}), y_i\right) \\
&\leq \sup_{i \in [n], \mathbf{x} \in \mathcal{B}(\mathbf{x}_i)} \ell\left(f(\mathbf{W}, \mathbf{x}), y_i\right) \\
&\leq L \sup_{i \in [n], \mathbf{x} \in \mathcal{B}(\mathbf{x}_i)} \left( |f(\mathbf{W}, \mathbf{x}) - g_2(\mathbf{x})| + |g_2(\mathbf{x}) - g_1(\mathbf{x})| + |g_1(\mathbf{x}) - y_i| \right) \\
&\leq 3L\bar{\epsilon} \\
&= \epsilon.
\end{aligned}
$$

The theorem follows by setting $\delta = 0.01$. $\qquad\qquad\square$

## C.2  Example of Using Quadratic ReLU Activation

Theorem 5.2 shows that when Assumption 5.2, 5.1, 5.3 hold, we can indeed find a classifier of low robust loss within a neighborhood of the initialization. However, these assumptions are either for generality or simplicity, and for specific activation functions, we can remove these assumptions. As a guide example, we consider the *quadratic ReLU* function $\sigma(x) = \mathrm{ReLU}(x)^2 = x^2 \cdot 1_{x \geq 0}$ and its induced NTK. Following the previous work [3, 2], we consider the initialization of each $\mathbf{w}_r$ with the *uniform distribution on the surface of the sphere of radius $\sqrt{d}$* in this section.[4] We can verify that this induced kernel is universal and quantitatively derive the dependency of $\epsilon$ for $R_{\mathcal{D}, \mathcal{B}, \epsilon}$ and $m$ in Theorem 5.2 for this two-layer network. In order to do so, we need to make a mild assumption of the dataset:[5]

**Assumption C.1** (Non-overlapping). *The dataset $\{\mathbf{x}_i, y_i\}_{i=1}^n \subset \mathcal{S}$ and the perturbation set function $\mathcal{B}$ satisfies the following: There does not exist $\mathbf{x}, \bar{\mathbf{x}}$ and $i, j$ such that $\mathbf{x} \in \overline{\mathcal{B}(\mathbf{x}_i)} \cup (-\overline{\mathcal{B}(\mathbf{x}_i)}), \bar{\mathbf{x}} \in \overline{\mathcal{B}(\mathbf{x}_j)} \cup (-\overline{\mathcal{B}(\mathbf{x}_j)})$ but $y_i \neq y_j$.*

And then we can derive the finite-sum approximation result by random features.

**Theorem C.1** (Approximation by Finite Sum). *For a given Lipschitz function $h \in \mathcal{H}(K_\sigma)$. For $\epsilon > 0, \delta \in (0, 1)$, let $\mathbf{w}_1, \cdots, \mathbf{w}_M$ be sampled i.i.d. from the uniform distribution on the surface of the sphere of radius $\sqrt{d}$ where*

$$M = \Omega \left( C_{\mathcal{D}, \mathcal{B}} \frac{1}{\epsilon^{d+1}} \log \frac{1}{\epsilon^{d+1}\delta} \right). \tag{12}$$

*and $C_{\mathcal{D}, \mathcal{B}}, C'_{\mathcal{D}, \mathcal{B}}$ is a constant that only depends on the dataset $\mathcal{D}$ and the compatible perturbation $\mathcal{B}$. Then with probability at least $1 - \delta$, there exists $c_1, \cdots, c_M$ where $c_i \in \mathbb{R}^d$ such that $\hat{h} = \sum_{r=1}^M c_r^\top \mathbf{x} \mathrm{ReLU}\left(\mathbf{w}_r^\top \mathbf{x}\right)$ satisfies*

$$\sum_{r=1}^M \|c_r\|_2^2 = O\left( \frac{C'_{\mathcal{D}, \mathcal{B}}}{M} \right), \tag{13}$$

$$\left\| h - \hat{h} \right\|_{\infty, \mathcal{S}} \leq \epsilon. \tag{14}$$

To prove this theorem, we use the $\ell_2$ approximation result in [3] and translate it to an $\ell_\infty$ approximation result by using Lipshitz continuity. We first state Proposition 1 in [3].

**Lemma C.2** (Approximation of unit ball of $\mathcal{H}(K_\sigma)$, Corollary of Proposition 1 in [3]). *Let* $h \in \mathcal{H}(K_\sigma)$. *For* $\epsilon > 0$, *let* $d\rho$ *be the uniform distribution on* $\mathcal{S}$. *Let* $\mathbf{w}_1, \cdots, \mathbf{w}_M$ *be sampled i.i.d. from uniform distribution on the surface of the sphere of radius* $\sqrt{d}$, *then for any* $\delta \in (0, 1)$, *if*

$$M = \exp(\Omega(d)) \frac{\|h\|_{\mathcal{H}}^2}{\epsilon} \log\left( \frac{\|h\|_{\mathcal{H}}^2}{\epsilon\delta} \right),$$

*with probability at least* $1 - \delta$, *there exists* $c_1, \cdots, c_M \in \mathbb{R}^d$ *such that* $\hat{h} = \sum_{r=1}^M c_r^\top \mathbf{x} \mathrm{ReLU}\left( \mathbf{w}_r^\top \mathbf{x} \right)$ *satisfies*

$$\sum_{r=1}^M \|c_r\|_2^2 = \frac{\|h\|_{\mathcal{H}}^2 \exp(O(d))}{M}, \tag{15}$$

$$\left\| h - \hat{h} \right\|_{L_2(d\rho)}^2 = \int_{\mathcal{S}} \left( h - \hat{h} \right)^2 d\rho \le \epsilon. \tag{16}$$

Then we can give the proof of Theorem C.1.

*Proof.* Let $Lip(f)$ denote the Lipschitz coefficient of $f$. We consider $\hat{h}$ in Lemma C.2, by the property of Lipschitz coefficient, we have

$$
\begin{aligned}
Lip(\hat{h}) =& Lip\left( \sum_{r=1}^M c_r^\top \mathbf{x} \mathrm{ReLU}\left( \mathbf{w}_r^\top \mathbf{x} \right) \right) \\
\le& \sum_{r=1}^M Lip\left( c_r^\top \mathbf{x} \mathrm{ReLU}\left( \mathbf{w}_r^\top \mathbf{x} \right) \right) \\
\le& \sum_{r=1}^M \|c_r\|_2 \|\mathbf{x}\|_2 \, Lip\left( \mathrm{ReLU}\left( \mathbf{w}_r^\top \mathbf{x} \right) \right) \\
\le& \sum_{r=1}^M \|c_r\|_2 \|\mathbf{w}_r\|_2 \\
\le& \sqrt{\left( \sum_{r=1}^M \|c_r\|_2^2 \right) \left( \sum_{r=1}^M \|\mathbf{w}_r\|_2^2 \right)},
\end{aligned}
$$

So

$$Lip(\hat{h}) = \|h\|_{\mathcal{H}} \exp(O(d)),$$

which means $\hat{h}$ has finite Lipschitz coefficient and therefore so does $h - \hat{h}$, and the upper bound of Lipschitz constant $c_L$ only depends on the data and the perturbation. Then we can bound the $\ell_\infty$ approximation error. Suppose for some $\mathbf{x} \in \mathcal{S}$, $\left| h(\mathbf{x}) - \hat{h}(\mathbf{x}) \right| > \epsilon$, since $h - \hat{h}$ is Lipschitz, it is not hard to see that, when $\epsilon$ is small,

$$\int_{\mathcal{S}} \left( h - \hat{h} \right)^2 \gtrsim \frac{\pi^{\frac{d}{2}} \epsilon^{d+1}}{\Gamma(d/2+1)c_L^d} \asymp \frac{\epsilon^{d+1}}{c_L^d} \frac{(2\pi e)^{\frac{d}{2}}}{d^{\frac{d+1}{2}}}. \tag{17}$$

By Lemma C.2, for some constant $C_{\mathcal{D},\mathcal{B}}$, when $M = \Omega\left( \frac{C_{\mathcal{D},\mathcal{B}}}{\epsilon^{d+1}} \log \frac{1}{\epsilon^{d+1}\delta} \right)$, Equation (17) fails, so $\left\| h - \hat{h} \right\|_{\infty,\mathcal{S}} \le \epsilon$ holds and at the same time we have $\sum_{r=1}^M \|c_r\|_2^2 = O\left( \frac{C'_{\mathcal{D},\mathcal{B}}}{M} \right)$ for some $C'_{\mathcal{D},\mathcal{B}}$ that only depends on the data and the perturbation. $\square$

Now, we get a similar but more explicit finite-sum approximation result for quadratic ReLU activation, we are then going to show that the RKHS is rich enough that Assumption 5.2 can be derived. We have the following lemma to characterize the capacity of the RKHS.

**Lemma C.3** (RKHS Contains Smooth Functions, Proposition 2 in [2], Corollary 6 in [4]). *Let $f : \mathcal{S} \to \mathbb{R}$ be an even function such that all $i$-th order derivatives exist and are bounded by $\eta$ for $0 \leq i \leq s$, with $s \geq (d+3)/2$. Then $f \in \mathcal{H}(K_\sigma)$ with $\|f\|_{\mathcal{H}} \leq C_d \eta$ where $C_d$ is a constant that only depend on the dimension $d$.*

Then, by plugging-in the finite-sum approximation theorem (Theorem C.1) and the theorem of the capacity of RKHS (Theorem C.3) to the proof of Theorem 5.2 and combining with the optimization theorem, we can get an overall theorem for the quadratic-ReLU network which is similar to Corollary 5.1 but with explicit $\epsilon$ dependence:

**Corollary C.1** (Adversarial Training Finds a Network of Small Robust Train Loss for Quadratic-ReLU Network). *Given data set on the unit sphere equipped with a compatible perturbation set function and an associated perturbation function $\mathcal{A}$, which also takes value on the unit sphere. Suppose Assumption 3.1, C.1 are satisfied. Let $C''_{\mathcal{D},\mathcal{B}}$ be a constant that only depends on the dataset $\mathcal{D}$ and perturbation $\mathcal{B}$. Then for any 2-layer quadratic-ReLU network with width $m = \Omega(\frac{C''_{\mathcal{D},\mathcal{B}}}{\epsilon^{d+1}} \log \frac{1}{\epsilon})$, if we run gradient descent with stepsize $\alpha = O(\epsilon)$ for $T = \Omega(\frac{\sqrt{m}}{\alpha})$ steps, then with probability $0.99$,*

$$\min_{t=1,\cdots,T} L_{\mathcal{A}}(\mathbf{W}_t) \leq \epsilon. \tag{18}$$

# D  Proof of Theorem 6.1

*Proof.* We prove this theorem by an explicit construction of $\left\lceil \frac{n}{2} \right\rceil \times d$ data points that $\mathcal{F}$ is guaranteed to be able to shatter. Consider the following data points

$$\mathbf{x}_{i,j} = \mathbf{c}_i + \epsilon \mathbf{e}_j \text{ for } i \in \left\{1, \cdots, \left\lceil \frac{n}{2} \right\rceil \right\}, j \in [d],$$

where $\mathbf{c}_i = (6i\delta, 0, \cdots, 0)^\top \in \mathbb{R}^d$, $\epsilon$ is a small constant, and $\mathbf{e}_j = (0, \cdots, 1, \cdots, 0) \in \mathbb{R}^d$ is the $j$-th unit vector. For any labeling $y_{i,j} \in \{1, -1\}$, we let $P_i = \{j \in [d] : y_{i,j} = 1\}$, $N_i = \{j \in [d] : y_{i,j} = -1\}$, and let $\#P_i = k_i$. The idea is that for every cluster of points $\{\mathbf{x}_{i,j}\}_{j=1}^n$, we use 2 disjoint balls with radius $\delta$ to separate the positive and negative data points. In fact, for every such cluster, if $P_i$ and $N_i$ are both non-empty, the hyperplane

$$\mathcal{M}_i = \{\mathbf{x} : (y_{i,1}, \cdots, y_{i,d}) \cdot (\mathbf{x} - \mathbf{c}_i) = 0\},$$

clearly separates the points into $\{\mathbf{x}_{i,j} : j \in P_i\}$ and $\{\mathbf{x}_{i,j} : j \in N_i\}$. Then we can see easily that there exists a $\gamma_{k_i} > 0$ such that for any $r > \gamma_{k_i}\epsilon$, there exist two Euclidean balls $\mathcal{B}_r(\mathbf{x}'_{i,1}), \mathcal{B}_r(\mathbf{x}'_{i,2})$ in $\mathbb{R}^d$ with radius $r$, such that they contain the set $\{\mathbf{x}_{i,j} : j \in P_i\}$ and $\{\mathbf{x}_{i,j} : j \in N_i\}$ respectively, and that $\mathcal{B}_r(\mathbf{x}'_{i,1})$ and $\mathcal{B}_r(\mathbf{x}'_{i,2})$ are also separated by $\mathcal{M}_i$. Therefore, as long as we take

$$\epsilon < \delta \max\left(\frac{1}{\gamma_1}, \cdots, \frac{1}{\gamma_{d-1}}, 1\right),$$

we can always put $r = \delta$. This also holds in the case that $P_i$ or $N_i$ is empty, where we can simply put one ball centered at $\mathbf{x}_{i,1} = \mathbf{c}_i$ and put $\mathcal{B}_r(\mathbf{x}_{i,2})$ anywhere far away so that it is disjoint from the other balls. Recall that we have chosen $\|\mathbf{c}_i - \mathbf{c}_{i'}\|_2 \geq 6\delta$ for $i \neq i'$. Such balls $\mathcal{B}_r(\mathbf{x}_{i,l}) : i \in \left\{1, \cdots, \left\lceil \frac{n}{2} \right\rceil \right\}, l \in \{1, 2\}$, are disjoint since $\epsilon \leq \delta$, and $\left\| \mathbf{x}'_{i,l} - \mathbf{c}_i \right\|_2 \leq 2\delta$ for $l = 1, 2$. In this way, since $\mathcal{F}$ is an $n$-robust interpolation class, we can use the fact that there exists a function $f \in \mathcal{F}$ such that for any $i \in \left\{1, \cdots, \left\lceil \frac{n}{2} \right\rceil \right\}$, $f(\mathbf{x}) = 1$ for $\mathbf{x} \in \mathcal{B}_r(\mathbf{x}_{i,1})$ and $f(\mathbf{x}) = -1$ for $\mathbf{x} \in \mathcal{B}_r(\mathbf{x}_{i,2})$. In this way, $f(\mathbf{x}_{i,j}) = y_{i,j}$ holds for all $i, j$. Since the labels $y_{i,j}$ can be picked at will, by the definition of the VC-dimension, we know that the VC-dimension of $\mathcal{F}$ is always at least $\left\lceil \frac{n}{2} \right\rceil \times d$. $\qquad \square$

## Footnotes

[1] Note that in [1] the corresponding region $B(R)$ is defined by the 2-norm instead of the $F$-norm: $B_2(R) := \{\mathbf{W} : \left\|\mathbf{W}^{(h)} - \mathbf{W}_0^{(h)}\right\|_2 \le \frac{R}{\sqrt{m}}, \forall h \in [H]\}$. Since obviously $B_2(R) \subset B(R)$, we can still apply their results to our case directly.

[2] Here the zero norm $\|\cdot\|_0$ denotes the number of non-zero entries of a matrix or a vector.

[3] We only use the setting when the network output is a scalar.

[4]It is not hard to see that Theorem 5.1 still holds under this situation by the same proof.

[5]Our assumption on the dataset essentially requires $\mathbf{x}_i \neq \pm \mathbf{x}_j$ since the quadratic ReLU NTK kernel only contains even functions. However, this can be enforced via a lifting trick: let $\tilde{\mathbf{x}} = [\mathbf{x}, 1] \in \mathbb{R}^{d+1}$, then the data $\tilde{\mathbf{x}}$ lie on the positive hemisphere. On the lifted space, even functions can separate any datapoints.