[Reviews · NeurIPS 2019]

Reviewer 1



(1) This paper has a very clear presentation. The overall writing is quite good. (2) The convergence proof in this paper is general --- no attacker is specified. Overall, the reviewer feels it is an important and nice theoretical work on adversarial training. This paper may shed lights on future works on designing better or understanding adversarial training. However, these theoretical proofs beyond the reviewer’s domain knowledge, the reviewer is unable to justify the correctness of these math proofs.

Reviewer 2



EDIT: I have read the author feedback and the authors have agreed to revise the writing. This is clearly a good paper that should be accepted. Two more comments regarding the rebuttal: (1) My original comments apply to natural training as well, and I understand this is a very challenging topic. (2) I understand one can reduce the dependence on H, but my concern was that probably more parameters are needed to more precisely characterize the landscape (for both natural and adversarial training). For example, one such parameter is width. -------------------- This is an original paper that develops convergence results (of training error) of adversarial training for extremely wide networks. As far as I know, this is also first such results for adversarial training. The high-level strategy of the proof follows the recent paradigm of analyzing extremely wide networks for natural training (more precisely, through NTK), where the behavior of the networks becomes almost linear, and thus is amenable to precise analysis. This is a good paper, I like it, and vote to accept. That being said, the current convergence theory is quite unsatisfying (I would say the same thing for the convergence theory we currently have for natural training). Among many things, I want to highlight two things that are particularly unsatisfying to me: 1. Depth of the networks *only* hurts convergence in this theory, and it is unavoidable. Let's start by observing that the current dependence on H is 2^{O(H)} (this is despite of the fact that the required width depends linearly on the number of training points, which is a separate thing), and moreover, as H increases, we will strictly need (exponentially) wider networks. This is very counter-intuitive, because empirically, theoretically (e.g., from the research of circuit complexity), and intuitively, H should at least sometimes help, rather than only hurting the convergence. Moreover, such a "monotonic dependence on H" is unavoidable because H is the *only* architectural parameter used by the theory to characterize convergence. And it is impossible to get rid of this monotonic dependence without injecting more parameters. For example, one could imagine the following theory by measuring one more parameter as a complexity measure: Suppose that we try to interpolate and recover certain hidden function f through data, and given depth H, the optimal neural network for representing f has width w_H(f). In this case, one could hope that for some good f, the width requirement for overparameterized networks to converge is a function of w_H(f), instead of H. Note that, w_H(f) could potentially be much smaller than H. For example, if we study f=XOR, then as H gets large enough, w_H(f) = O(1), and is independent of H, and so, even if we have a theory where the overparameterized network is of width 2^{w_H(XOR)}, it is a constant for large H (2^{w_H(f)} = 2^{O(1)} = O(1)), and thus give huge improvement. There are many things that can be explored in such an extended theory -- and I believe that that will lead to a much more reasonable theory. 2. From this theory, what are the "testable hypotheses" for guiding empirical work on neural networks? I think a downside with the current "foundations of deep learning" research is that it is more like "explanation for explanation's purpose", and lacks "predictions". I believe that a good theory should be able to predict something useful and guide the practice (which is critical and common in say a physics theory). Yet, I am not sure what the current theory can say about this ( maybe there is but it is not written in the paper).

Reviewer 3



EDIT: I have read the author feedback, and the authors have explicitly discussed the (minor) issues I raised. They have agreed to clarify the wording of heuristic v. exact inner maximization, and be more forthright about the success of adversarial training having to do with the expressivity of the neural tangent kernel. They have even gone beyond what I expected and discussed removing the projection step for two-layer networks. As such, I have increased my score from a 7 to an 8. I believe this is a strong paper. -------------- The paper is to the best of my knowledge, wholly original. The work ties together many important theoretical tools that have been developed over the last year (overparameterized neural networks, neural tangent kernels, adversarial robustness, etc.) and ties them together in an important and interesting way. Moreover, this is the first theoretical result (to my knowledge) that supports the idea of using adversarial training to find robust neural networks, amplifying its originality. The paper is very well-written. The authors do a good job of detailing their contributions, connecting it to related work, and giving proof sketches. The theorems seem to be technically sound. The authors are generally honest about the drawbacks of their theory. One issue I have here is that the authors purport to show multiple times that neural networks can find robust classifiers. However, this holds only under assumption 5.1. Due to the complexity of the neural tangent kernel, it is not clear whether this assumption is likely to hold. I would argue that the authors should be forward facing about this in the introduction, in the interest of transparency. The clarity of the work is generally very high. The paper is well-organized, as are the proofs. I believe the paper should also be commended for taking relatively modern theory on the convergence of gradient descent for neural networks, and managing to simplify and expand upon this work, in a way that is eminently comprehensible. My one issue with clarity is that some of the contributions of the paper could be expressed in clearer ways. For example, the second contribution (end of page 1, beginning of page 2) refers to a "data dependent RF norm", which is not explained till much later. Another somewhat confusing part is in the abstract, which differentiates "exact" and "heuristic" solving of the inner maximization of adversarial training. However, this distinction can be omitted for clarity, as is done in the first bullet point contribution, which I personally think would benefit the paper. I do not think this is necessary by any means, it is simply a suggestion I wanted to make. The paper's significance could be subject to some debate. It is not directly useful at a practical level, except in the sense that it gives some evidence towards the efficacy of adversarial training. Moreover, even the formal theorems of the paper require assumptions that are not met in practice, such as having exponentially wide networks. However, I would argue that this paper is still significant. First, quantifying even the training error, much less the robust training error, of a neural network is difficult, and this paper is able to make progress on this. Similarly, the paper implicitly develops techniques that can aid other theoretical analyses of neural networks. The work could help other adversarial training papers, as it expands the traditional scope of adversarial training (to so-called perturbation sets and functions) and shows that these more general adversarial training methods can still be analyzed. The paper also helps expand upon recent and exciting work on neural tangent kernels. Finally, it makes simple but important connections between VC dimension and adversarial robustness, which again can help inspire more theory. The above reads like a laundry list of various theoretical tools developed for understanding machine learning theoretically, and I believe the paper is significant for tying these all together.

[Author Response · NeurIPS 2019]

We thank the reviewers for their thorough and thoughtful reviews. We greatly appreciate the positive comments and address the questions below.

**To Reviewer #1:** We thank the reviewer for the comments.

(1) Although there has been no theoretical guarantees before, the convergence of adversarial training to zero loss is well-observed in practice. There are papers, e.g. Madry et al. [24], showing that as the capacity of network increases, adversarial training will converge to nearly zero loss. Moreover, we also conducted experiments showing the convergence of adversarial training for different architectures. For the 3x-wide and 10x-wide Resnet-32 (solid red and green lines), the training accuracy is close to 100%.

Figure 1: Adversarial Training with Different Architectures. $y$-axis is accuracy and $x$-axis is epochs.

(2) Generalization in the robustness literature is an important problem that is not addressed in this paper. We will add a discussion of the work on robust generalization which is complementary to this paper. In future work, we plan on investigating how our adversarial training results can be combined with robust generalization to yield end-to-end guarantees on the robust test loss.

(3) Thank you for pointing out the two papers related to the capacity argument; we will cite these in the next version and discuss the relationship.

**To Reviewer #2:** We thank reviewer 2 for giving insightful suggestions on both theory and writing.

We wholeheartedly agree that we should talk more about the limitations of the current theory and point out the future directions more clearly. Some of this discussion is in Section 7, especially the possibility of reducing the exponential dependence of the depth into polynomial dependence (we believe using similar techniques in reference [1], reducing to polynomial dependence is indeed possible without changing the structure of the arguments in this paper and potentially even only logarithmic depth dependence via using a ResNet architecture). We will discuss in more detail in the revision, including the need for more fine-grained analysis on the role of depth, architecture, and input data. Of course even for natural training, many of these questions remain open. We will expand upon Section 7 the limitations of the current analysis and routes to improve the analysis, and finally testable hypotheses (stronger attack algorithm leads to stronger adversarial training loss guarantee and adversarial training requires additional capacity even to minimize the training loss).

**To Reviewer #3:** We thank reviewer 3 for the positive comments, and for giving insightful suggestions on both writing and future directions.

We will follow the reviewer's suggestions in the revision. In particular we will mention early on that the success of adversarial training is dependent upon the ability of the kernel method's expressivity. We will also try to reword the abstract to remove the ambiguity caused by exact vs heuristic inner maximization solving.

In addition, we are currently working on removing the projection in the gradient descent algorithm. For example, we can prove that for the two-layer case the projection step is not needed as remarked under Theorem 4.1; we will include the proof of this in the next version.



[Meta-Review · NeurIPS 2019]

The reviewers agreed the contributions made in this submission are significant and they all recommended acceptance.